# Multidimensional heuristic process for high-yield production of astaxanthin and fragrance molecules in *Escherichia coli*

Congqiang Zhang[1], Vui Yin Seow [2], Xixian Chen[1] & Heng-Phon Too [1,2]

Optimization of metabolic pathways consisting of large number of genes is challenging. Multivariate modular methods (MMMs) are currently available solutions, in which reduced regulatory complexities are achieved by grouping multiple genes into modules. However, these methods work well for balancing the inter-modules but not intra-modules. In addition, application of MMMs to the 15-step heterologous route of astaxanthin biosynthesis has met with limited success. Here, we expand the solution space of MMMs and develop a multidimensional heuristic process (MHP). MHP can simultaneously balance different modules by varying promoter strength and coordinating intra-module activities by using ribosome binding sites (RBSs) and enzyme variants. Consequently, MHP increases enantipure 3S,3'S-astaxanthin production to 184 mg l$^{-1}$ day$^{-1}$ or 320 mg l$^{-1}$. Similarly, MHP improves the yields of nerolidol and linalool. MHP may be useful for optimizing other complex biochemical pathways.

[1] Biotransformation Innovation Platform (BioTrans), Agency for Science, Technology and Research (A*STAR), Singapore 117597, Singapore. [2] Department of Biochemistry, National University of Singapore, Singapore 117597, Singapore. Correspondence and requests for materials should be addressed to C.Z. (email: zcqsimon@outlook.com) or to H.-P.T. (email: bchtoohp@nus.edu.sg)

Astaxanthin (3,3′-dihydroxy-β,β-carotene-4,4′-dione) is a xanthophyll carotenoid with multiple beneficial effects on human health. It exhibits strong antioxidant[1], anti-inflammatory[2], and anti-cancer activity[3]. As an approved oral dietary supplement, it prevents atherosclerotic cardiovascular disease[4] and diabetes[5]. Therefore, the demand for astaxanthin is growing rapidly in food, animal feed, nutraceuticals, cosmetics, and the pharmaceutical industry[6]. Currently, commercial astaxanthin is mainly obtained from extraction of natural producers (such as *Haematococcus pluvialis* algae) or through chemical synthesis[6]. However, the former route of production is very costly (>$7000 per kg), while the latter is not sustainable and poses concerns in health and food safety[7]. Alternatively, complete biosynthesis of astaxanthin from inexpensive carbon sources using microbes has the potential to increase yield and sustainability, thus lowering cost.

Astaxanthin has been producing at low levels in metabolically engineered *Escherichia coli*[8–12], *Saccharomyces cerevisiae*[13–15], and *Corynebacterium glutamicum*[16]. These low yields may be due to a poorly optimized isoprenoid pathway that diverts the carbon flux from glycolysis into carotenoids, or that the relative expression of downstream carotenoid biosynthetic pathway genes was not well controlled[9,10]. Starting from acetyl-CoA, the astaxanthin biosynthetic pathway proceeds linearly until β-carotene, from which the pathway continues via complex multiple routes[10]. Several groups have successfully engineered microbes to produce high levels of lycopene and β-carotene[17–21], but not astaxanthin[9,10]. Hence, a significant challenge is to find ways to optimize the pathway where there are 15 distinct biochemical reactions from acetyl-CoA to astaxanthin, and to finely tune the final steps, which are catalyzed by hydroxylases and ketolases.

Pathway design and optimization are essential in the commercial production of chemicals by microbial factories[22]. Two major strategies are used to optimize metabolic pathways. One is to investigate the full combinatorial libraries of each gene with different regulatory elements (e.g., promoters, RBSs). This is comprehensive but ineffective in cost of time and resources as the construction and screening of libraries increase exponentially with increasing number of genes[23]. The other strategy is to group genes into modules and control the expression by tuning regulatory elements such as promoters. An example is the multivariate modular metabolic engineering (MMME) approach[24], which elegantly segments metabolic pathways into modules regulated by distinct promoters. When involving metabolic pathways with large numbers of genes, MMME approaches will require the exploration of a large number of possible combinations of modules. To mitigate this challenge, experimental designs (e.g., fractional factorial design) and regression models are used to pinpoint globally optimal phenotypes and identify contributions from the various modules and abiotic conditions[25–27]. Regardless, the issue of imbalance of enzymatic activities in any one module resulting in poor production remains to be identified and resolved. Attempts to model enzymatic kinetics in vivo have met with limited success, thereby making predictions of the contributions of enzymatic activities in the modules challenging[28].

Herein, a multidimensional heuristic process (MHP) is developed to address the limited dimensional spaces of current modular approaches. MHP is a modular pathway optimization process to generate high producer strains by assembling and screening diverse libraries of predefined regulatory elements together with key enzyme variants in a high dimensional combinatorial manner. One dimension involves the construction of

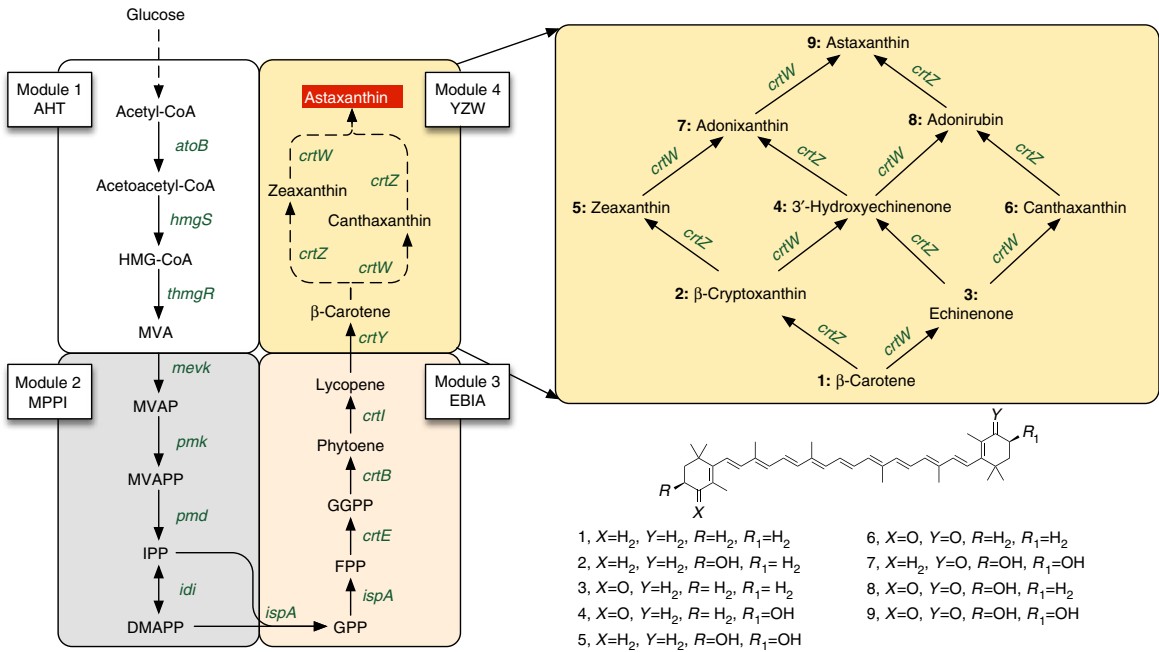

**Fig. 1** Biosynthetic pathway of astaxanthin. The biosynthetic pathway was grouped into four major modules: the upstream MVA pathway (module 1 or AHT, including *atoB*, *hmgS*, and *thmgR*), the downstream MVA pathway (module 2 or MPPI, including *mevk*, *pmk*, *pmd*, and *idi*), the lycopene pathway (module 3 or EBIA, including *crtEBI* and *ispA*) and the astaxanthin pathway (module 4 or YZW, including *crtYZW*). Abbreviation for the compounds: HMG-CoA, 3-hydroxy-3-methyl-glutaryl-coenzyme A; MVA, mevalonate; MVAP, phosphomevalonate; MVAPP, diphosphomevalonate; IPP, isopentenyl pyrophosphate; DMAPP, dimethylallyl pyrophosphate; GPP, geranyl pyrophosphate; FPP, farnesyl pyrophosphate; GGPP, geranylgeranyl pyrophosphate. Dashed arrow indicates multiple enzymatic steps. The genes expressed encode the following enzymes: *atoB*, acetoacetyl-CoA thiolase; *hmgS*, HMG-CoA synthase; *thmgR*, truncated HMG-CoA reductase; *mevk*, mevalonate kinase; *pmk*, phosphomevalonate kinase; *pmd*, mevalonate pyrophosphate decarboxylase; *idi*, IPP isomerase; *ispA*, FPP synthase; *crtE*, GGPP synthase; *crtB*, phytoene synthase; *crtI*, phytoene desaturase; *crtY*, lycopene cyclase; *crtW*, β-carotene ketolase; *crtZ*, β-carotene hydroxylase

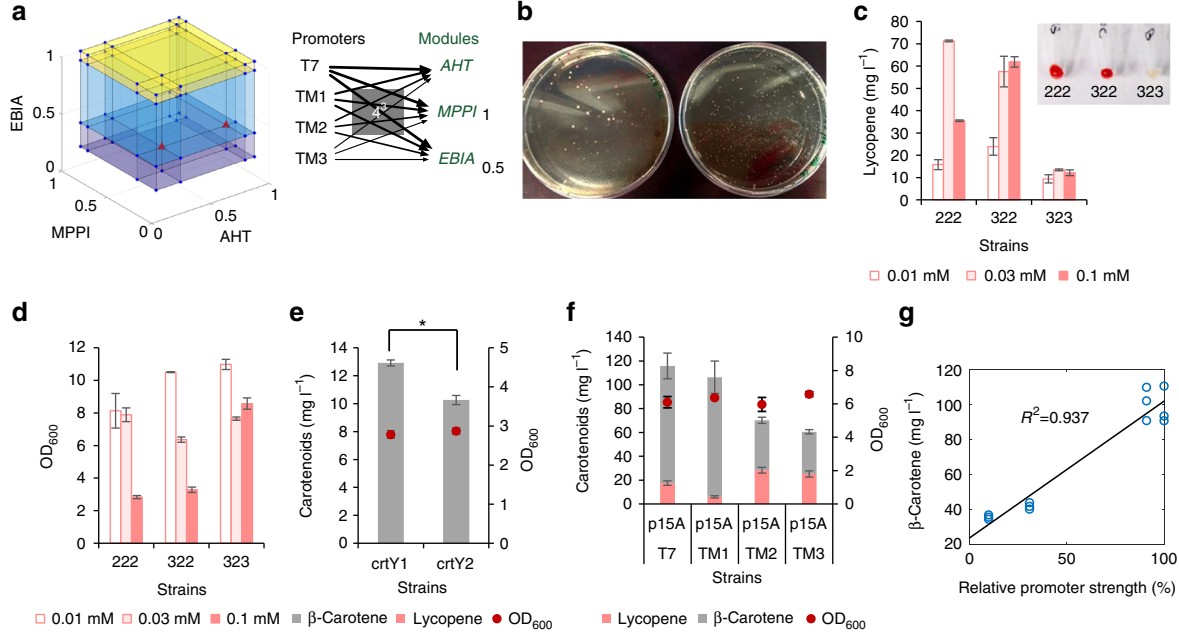

**Fig. 2** Sequential optimization of different carotenoids. The promoter library (T7, TM1, TM2 and TM3) was used to optimize the mevalonate pathway and carotenoid production. **a** The whole pathway (the mevalonate pathway and lycopene biosynthetic pathway) was divided into 3 modules (AHT, MPPI and EBIA in Fig. 1). Each module was controlled by 4 different promoters, thus obtaining 64 different combinations. **b** Agar plates with 0.1 mM IPTG, we selected distinguished phenotypes (colonies with intense red color) and reversed engineered the colonies. Validation of the chosen genotypes in liquid media: **c** lycopene titers and **d** cell density, $OD_{600}$. On top of lycopene strain 322, we first compared two different lycopene cyclases (crtY). **e** The cyclase from *Pantoea ananatis* LMG20103 (crtY1) performed better than that from *Uncultured marine bacterium* HF10_19P19 (crtY2). **f** Improving β-carotene production by tuning the promoter strength of crtY1. **g** Correlation between β-carotene production and the promoter strength. Error bars, mean ± s.d., $n = 3$. Statistically significant difference of carotenoids produced by two cyclases was denoted *$P < 0.05$ (two-tailed Student's t-test)

promoters of different strength, globally controlling the transcription of modules but not the relative expression of intra-modular genes. Another dimension is the construction of variable 5′ untranslated regions (5′UTR), functioning as the local regulators controlling single-gene translation in a specific module. Lastly, a dimension involving enzyme variants either from various organisms or from enzyme engineering technology (e.g., by DNA shuffling) is introduced as these variants often have significantly different and unpredictable expressions, activities and substrate selectivity. To demonstrate the utility of MHP, we applied it to optimize the production of three distinct terpenoids (one deeply colored compound, 3S,3′S-astaxanthin, and two others, linalool and nerolidol).

## Results

**Optimizing the production of lycopene and β-carotene.** There were two options to optimize the production of astaxanthin. One was to simultaneously control all the 14 genes, as shown in Fig. 1. However, this option contained a relatively large number of combinations thus rendered the construction and screening difficult. More importantly, it was difficult to diagnose the bottleneck in the system. Instead, we chose to sequentially optimize the biosynthetic pathways by overproducing lycopene, β-carotene and astaxanthin stepwise. Here, we chose lycopene, the first red colored pathway intermediate and hence easily monitored by standard high-throughput colorimetric assays, as the metabolic hub to optimize the mevalonate and carotenoid pathway.

Briefly, we segmented the 14 genes into four modules (the genes *atoB*, *hmgS*, and truncated *hmgR* as module AHT, the genes *mevK*, *pmK*, *pmd*, and *idi* as module MPPI, the gene *crtEBI* and *ispA* as module EBIA and *crtY*, *crtZ*, and *crtW* as module

YZW, Fig. 1). The lycopene production was optimized by controlling the first three modules. Each module was expressed in a vector and controlled by four different promoters with pre-determined strength. After the cloning steps, we obtained a mixture of strains containing all the 64 combinations (Fig. 2a) and these were plated onto agar plates with inducers. We picked the 10 darkest red colonies as high lycopene producers and 1 white colony serving as a control (Fig. 2b). The red colonies belonged to two major combinations of T7 promoter variants (strain 322 and 222, summarized in Supplementary Table 1) and the white strain of promoter variant 323. We then further validated the strain performance in liquid medium. Strain 222, 322, and 323 produced up to 71.3 (or 46.1 mg g$^{-1}$ Dry Cell Weight (DCW)), 61.9 and 13.5 mg l$^{-1}$ of lycopene, respectively, Fig. 2c. The cell density of strain 322 and 222 was lower when higher amount of IPTG (0.1 mM) was used, but not for the control strain 323 (Fig. 2d), possibly due to the toxicity of high-content of lycopene in *E. coli* cells.

Building on lycopene strain 322, we further optimized the production of β-carotene by separately introducing two lycopene cyclases (crtY) from *Pantoea ananatis* (crtY1) and from an uncultured marine bacterium (crtY2) into a fourth vector (Fig. 1). The expression of *crtY1* resulted in a higher β-carotene content (12.9 mg l$^{-1}$) than that of *crtY2* (10.3 mg l$^{-1}$; Fig. 2e), indicating *crtY1* as a better choice for production. We then optimized the β-carotene production by tuning the expression of *crtY1* with four different promoters. The β-carotene production correlated almost linearly to the promoter strength (or transcriptional level of *crtY*, Supplementary Fig. 1), with the highest yield of 100.3 mg l$^{-1}$, or 47.0 mg g$^{-1}$ DCW (Fig. 2f and g). It is worthy to note that the linear correlation between production and promoter strength is the foundation for many mathematical models (For example,

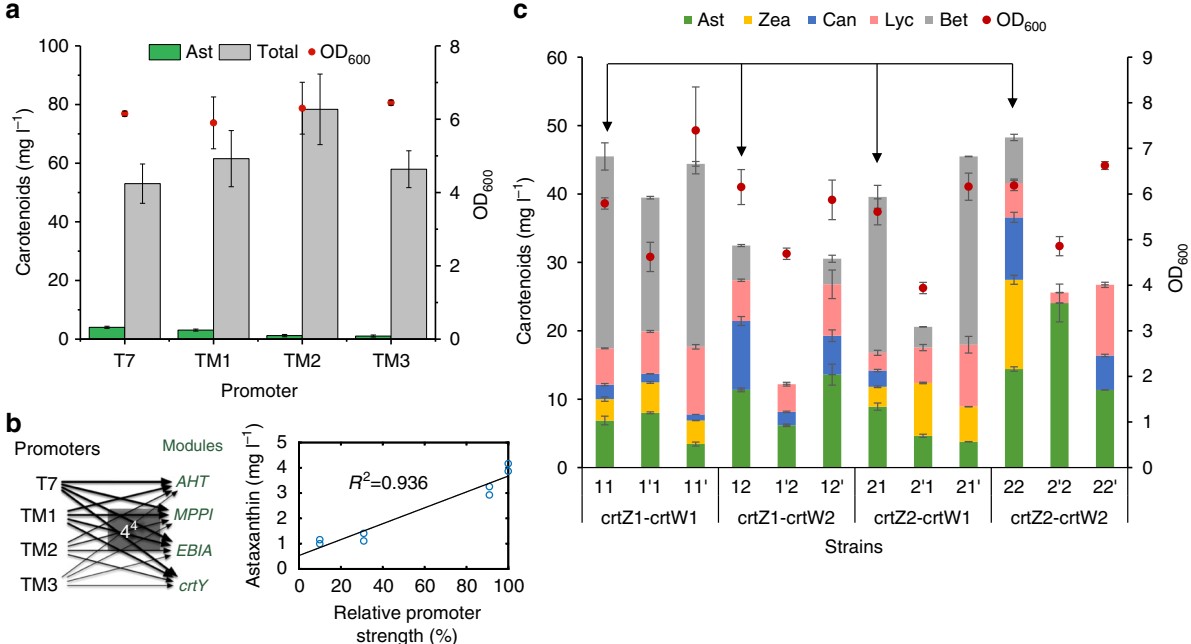

**Fig. 3** Optimization of astaxanthin production. **a** Global optimization for astaxanthin production using promoters, which was performed based on the optimized lycopene and β-carotene strain (Figs. 2 and 3) by fine tuning the transcription of the YZW module. **b** Correlation between astaxanthin production and the promoter strength that controls the transcription of the three genes, crtY, crtZ and crtW in the YZW module. **c** The local optimization of astaxanthin by investigating different β-carotene hydroxylases (crtZ) and ketolases (crtW) and translational control of individual enzyme. Strains 11, 12, 21, and 22 stand for the combination of crtZ1 and crtW1, crtZ1 and crtW2, crtZ2 and crtW1 and crtZ2 and crtW2, respectively. The symbol (′) in the figure refers to a different RBS, for example, 1′1 refers to the crtZ1 with a different RBS and 11′ refers to crtW1 with a different RBS (Supplementary Table 1 and 2). Error bars, mean ± s. d., n = 2. Arrows refer to the comparison among the four different combinations of enzymes (no RBS effects), or strains 11, 12, 21, and 22. Ast astaxanthin, Zea zeaxanthin, Can canthaxanthin, Lyc lycopene, Bet β-carotene, Total, all the five carotenoids

experimental design-aided systematic pathway optimization method[25]). The achieved lycopene strain and β-carotene strains have comparable or slightly better performances than previously reported[17–21].

**MHP markedly improved astaxanthin production**. Similar to the β-carotene work, the tuning of promoter for the fourth module YZW (Fig. 1) increased astaxanthin production (Fig. 3a), and there was an approximately linear correlation between promoter strength and the astaxanthin production (Fig. 3b). However, even with the strongest T7 promoter, astaxanthin production remained low with many other carotenoids accumulated in E. coli cells. All these global tuning approaches, where the three genes crtY, crtZ and crtW were tuned together by promoter or vector engineering, failed to further improve the astaxanthin yield. We hypothesized that the inability may be due to a suboptimal flux through the local astaxanthin pathway (Fig. 1). This hypothesis was supported by the observation that carbon fluxes were blocked at carotenoid intermediates (Fig. 3a). The next challenge was to optimize the astaxanthin pathway locally where there are six potentially different routes and seven possible intermediates from β-carotene to astaxanthin (Fig. 1). Previous attempts to enhance astaxanthin production by either using different ketolases or hydroxylases from various species have met with limited success[9,10]. To better control the metabolic fluxes, we next examined the effects of the expressions of the combinations of 2 different ketolases and 2 different hydroxylases using two RBSs with different strength (Supplementary Table 1 and 2). Among the four combinations (strain 11, 12, 21, and 22, Supplementary Table 1), strain 22 accumulated the highest yield of astaxanthin (14.4 mg l⁻¹) and zeaxanthin (13.0 mg l⁻¹, strains

with arrows, Fig. 3c). Strain 12 produced the highest yield of canthaxanthin (10.1 mg l⁻¹) and strain 11 accumulated the highest yield of β-carotene (28.1 mg l⁻¹). Consistent with the hypothesis, the choices of RBSs of ketolase and hydroxylase have significant effects on the production of astaxanthin and total carotenoids (Fig. 3c). For example, compared to strain 22, crtZ2 RBS mutation (strain 2′2) increased astaxanthin titer (24.1 mg l⁻¹) and purity (3–93% of total carotenoids, Fig. 3c and Supplementary Fig. 2). However, crtW2 RBS mutation (strain 22′) decreased the productions of astaxanthin (14.4–11.3 mg l⁻¹) and total carotenoids (48.3–26.7 mg l⁻¹).

The results in Fig. 3 demonstrated that, to achieve better yield and purity of astaxanthin, it was critical to optimize the relative expression of β-carotene ketolase and hydroxylase using RBS of different strength. Using astaxanthin as an example, the underlying principle and workflow of MHP were illustrated (Fig. 4) and demonstrated in the production of astaxanthin by manipulating the ketolation and hydroxylation steps. First, we designed RBS libraries using degenerate oligos for the control of crtZ and crtW, covering a 50-fold and a 140-fold range of relative RBS strength, respectively (Supplementary Fig. 3. Predicted RBS strength was listed in Supplementary Table 3 and 4). We then validated the experimental coverage of the RBS libraries by randomly picking 50 colonies from Library I and II. The 50 colonies represented ~90% of the designed space for the RBS of both crtZ and crtW (Supplementary Fig. 4). Subsequently, we optimized astaxanthin production by selecting strains from either single RBS library for crtZ, crtW, or with both RBS libraries simultaneously (Library I, II, and III in Supplementary Fig. 3, respectively). By plating the randomly cloned strains, a series of plates with strains accumulating different amounts of carotenoids were obtained (Fig. 5a). High astaxanthin accumulating cells were red in color,

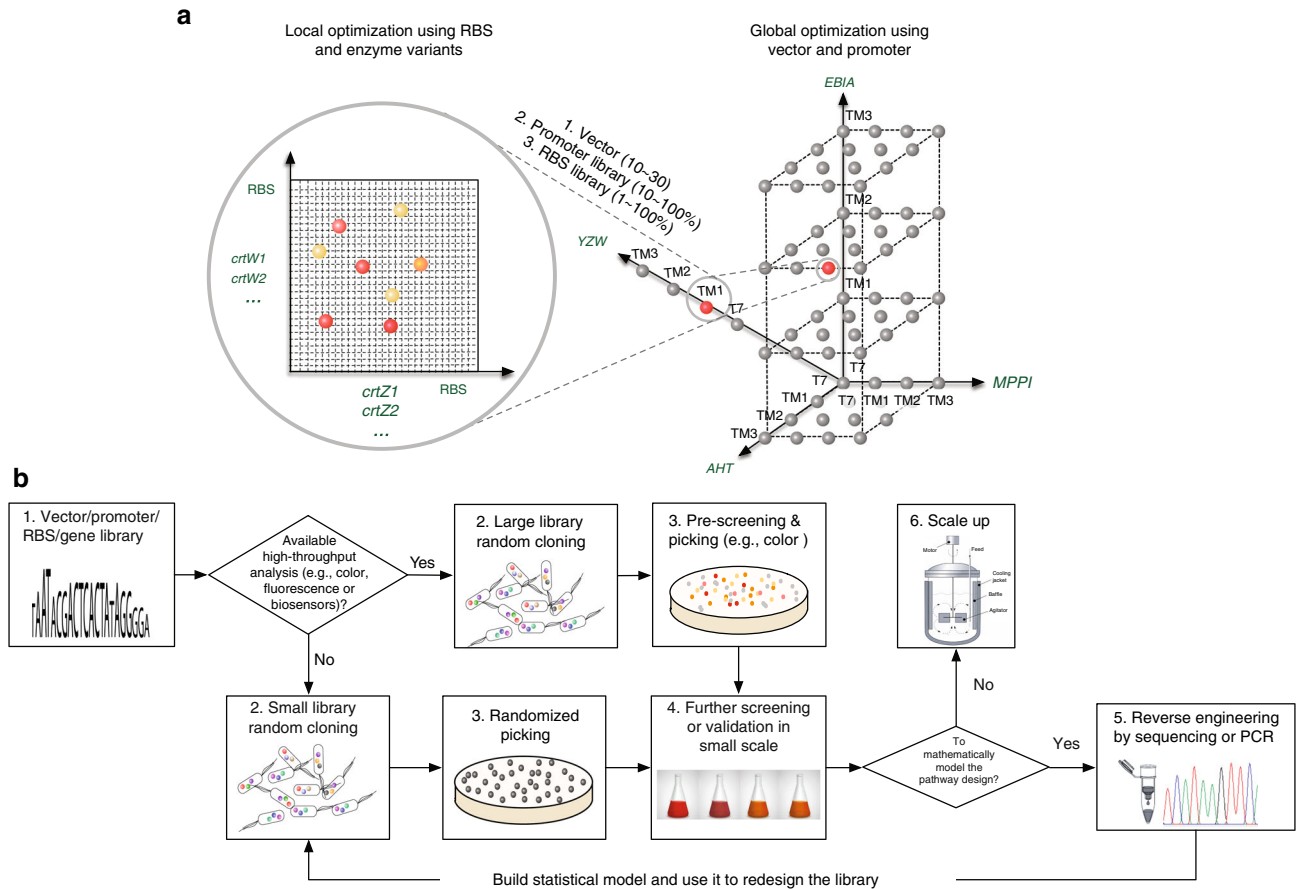

**Fig. 4** Illustration of MHP and its workflow. **a** In MHP, pathway engineering proceeds either sequentially or simultaneously with global optimization (promoters) and local refinement (5′UTRS or RBSs and enzyme variants). Globally, each promoter regulates the transcription of each operon/module with multiple genes (such as module AHT, MPPI, EBIA and YZW). Locally, various RBSs and enzymes facilitate fine tuning of the relative activities within one specific module (such as *crtW* and *crtZ* gene in YZW module). **b** The MHP workflow. First, libraries of promoters and RBSs were established and characterized either experimentally or mathematically. It was followed by a decision node that depends on the availability of high-throughput analytical methods of products (colorimetry or fluorescence). If so, the subsequent randomized cloning proceeded with a large combinatorial library ($10^{10}$–$10^{15}$) and the resulting strain library were pre-screened by colony color or fluorescence. Otherwise, the cloning continued with a small combinatorial library ($10^2$–$10^3$), followed by randomized picking and analyzed by low/medium throughput analysis (gas chromatography and liquid chromatography, etc). Next, performances of the strain were validated in liquid culture and desired phenotypes were further reversely engineered. The data could be then used to build statistical models and to refine new library to further improve phenotypes. Lastly, the optimized strain will be scaled up in bioreactors

which were distinct from the parental yellow β-carotene cells. We picked 5 red colonies and 1 yellow colony (control 1, Z2) from library I, 6 red colonies from library II and 23 red colonies and 1 yellow colony (control 2, ZW12) colonies from library III. Of the 36 strains (colonies) chosen, 19 of them (Z3-6, W4-6 and ZW13-24) were false positives with either low production or low purity of astaxanthin. The rest of the 17 strains (colonies) included 13 strains (ZW1-8, ZW11, Z1, and W1-3) which accumulated astaxanthin as the major carotenoid, 2 control strains (Z2 and ZW12) which produced β-carotene as the major carotenoid and 2 strains (ZW9-10) which accumulated astaxanthin and β-carotene as the major carotenoids (Fig. 5b). The total amount of carotenoids produced varied substantially among the different strains. Strain Z1 produced the highest concentration of astaxanthin, 53.5 mg l$^{-1}$ or 10 mg g$^{-1}$ DCW, i.e.,5-fold higher than that of the best global pathway optimized strain (Fig. 3a). We attempted to model the astaxanthin production by comparing the relative translation rates of *crtZ* and *crtW*, but there was no clear correlation (Fig. 5c, d). Hence, it appeared that there were yet to be characterized factors beyond merely regulating the relative translational initiation rates of β-carotene ketolase and hydroxylase.

**Extracellular carotenoids and its application.** Although colony color-based screening was an attractive approach to select colonies producing astaxanthin, there were many false positives due to difficulties in visual discrimination (19 out of 34 ≈62% of strains produced little or less pure astaxanthin). One confounding factor was the accumulation of other red colored compounds, such as lycopene (ZW2, W4 and ZW15) or canthaxanthin (ZW9 and Z6, Fig. 5b). The accumulation of zeaxanthin (yellow) and β-carotene (yellow) was less important, especially when comparing absorbance at 515 nm (Supplementary Fig. 5a). The challenge then was to develop a more specific selection approach that can distinguish astaxanthin from the rest of the major red colored intermediates. Interestingly, astaxanthin but not the other carotenoids can efficiently diffuse through the cell membranes (such as blood–brain barriers[29,30]) and this was partly due to the unique polar-nonpolar-polar structure that fits well into the cell membrane[31,32]. From these studies, we hypothesized that it might be possible to detect extracellular astaxanthin preferentially. If so, extracellular carotenoids (color in the growth media) would be a better screening indicator than intracellular carotenoids (color of the colonies).

To test this hypothesis, we compared the color of the growth media of the 43 different strains including the strains in Fig. 5 and

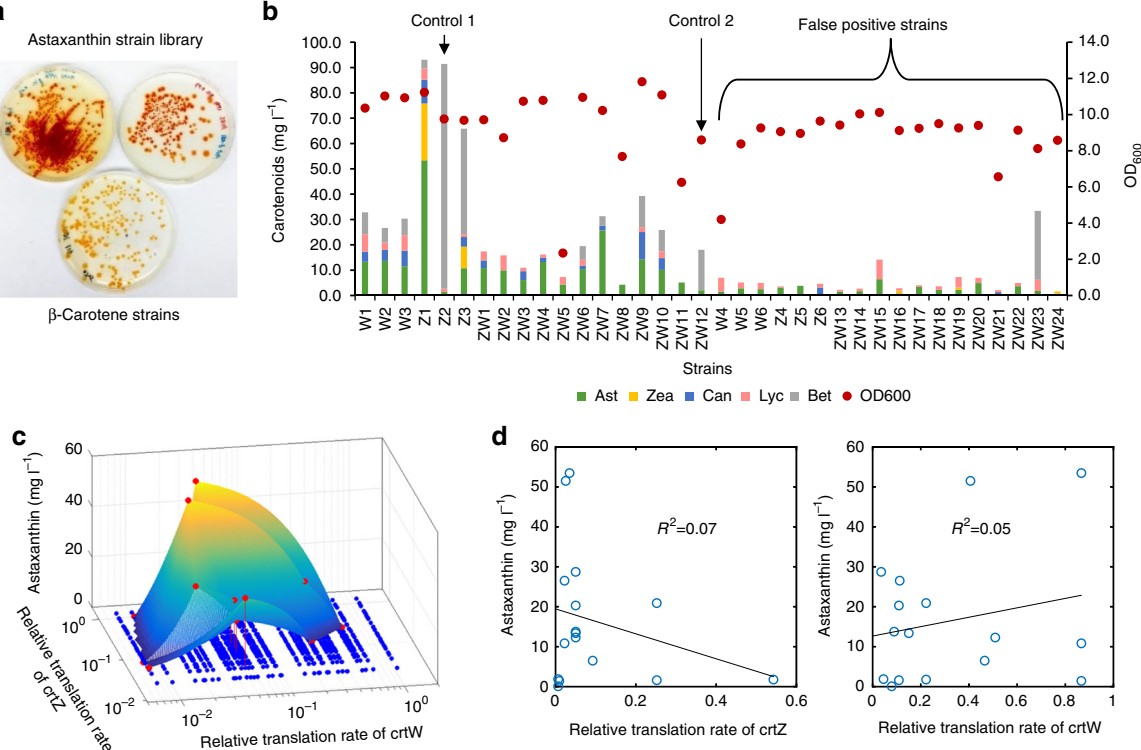

**Fig. 5** Improving astaxanthin production using MHP. **a** Agar plates of astaxanthin strains (the two plates above were different astaxanthin strains) and parental β-carotene strains (the plate below with yellow colonies). **b** Titers and cell density of different strains chosen from library I, II, and III. Strains Z1–Z6 were picked colonies from library I (*crtZ* RBS library), where Z2 was a yellow colony (control 1) and others were red colonies. Strains W1–W6 were picked red colonies from library II (*crtW* RBS library). Strains ZW 1–24 were picked colonies from library III (*crtZ* and *crtW* RBS libraries), where ZW12 was a yellow colony (control 2) and the rest were red colonies. **c** The 3-dimensional correlation of astaxanthin production with the translation rates of *crtZ* and *crtW*. **d** Scatterplot of astaxanthin production and RBS strength of *crtZ* or *crtW*

a few other strains from library I, II and III. Among the five carotenoids, the intracellular production of astaxanthin had the best correlation with absorbance at 515 nm (A515) of the growth media (Fig. 6a). In addition, both biomass and the confounding red intermediates, lycopene and canthaxanthin, had no correlation with A515 of the media. Furthermore, extracellular astaxanthin concentration (extracted and measured by HPLC) has a linear correlation with A515 of the media (Supplementary Fig. 5b) and the media has a very low background absorbance at 515 nm (Supplementary Fig. 5c). In addition, consistent with previous reports[29,30], our results supported the notion that astaxanthin has a superior transmembrane diffusional capability than other carotenoids. For example, despite producing high content of β-carotene in strain Z2 and Z3, there was no extracellular carotenoids detected (Supplementary Fig. 6 a,-c). In contrast, in strain Z1, the titers of extracellular astaxanthin, zeaxanthin and canthaxanthin were 15, 5, and 1 mg l$^{-1}$, respectively, accounting for 16%, 13%, and 8% of their total production, respectively (Supplementary Fig. 6d).

These results confirmed our hypothesis that colorimetry of culture media (measured at 515 nm) was useful for the screening of high astaxanthin-producing strains as astaxanthin diffuses out of the cells better than other carotenoids. We thus picked another 100 red colonies from the library III (Supplementary Fig. 3). Within the 100 chosen strains, we further selected two strains (E and F) with highest A515. Indeed, the two strains produced astaxanthin as the major product, 83% and 72% of total carotenoids, respectively, Fig. 6b and c. In addition, the yields of the two strains (~62 mg l$^{-1}$, 15.1 mg g$^{-1}$ DCW) were higher than all the other strains in this study and previous reports

(Supplementary Table 5). The results confirmed that colorimetry of culture media (or extracellular astaxanthin) was a more efficient screening approach for astaxanthin biosynthesis than the common colony screening.

In sum, the global control of metabolic pathways by promoter engineering increased astaxanthin yield from 1.0 to 4.0 mg l$^{-1}$ (or 0.4 to 1.9 mg g$^{-1}$ DCW) and purity from 2% to 8%. The subsequent introduction of two other local dimensions, the translational and the enzymatic dimension (β-carotene hydro-xylase and ketolase variants) resulted in further increase in astaxanthin titer by sevenfold (from 4.0 to 62 mg l$^{-1}$) and purity by tenfold from 8% to 82%.

**Fermentation testing and process development**. Using the best astaxanthin strain (strain E in Fig. 6b), we next explored the batch performance in a bioreactor using a chemically defined medium. Within 30 h, cell density reached an OD of 17, accumulated ~55 mg l$^{-1}$ astaxanthin (including 40 mg l$^{-1}$ intracellular and 15 mg l$^{-1}$ extracellular, Supplementary Fig. 7a). Consistent with tube and shake flask data, astaxanthin was the major product (~85% of total carotenoids) and there was insignificant amount of acetate in the fermentation broth. In order to improve production, we tested fed-batch process with stepwise feeding to maintain glu-cose level within the range of 0-15 g l$^{-1}$ (as described in meth-ods). When induction was carried out at low cell density (an OD of 10), a relatively low biomass was achieved (final OD was only 36 and about 3.5 g l$^{-1}$ acetate was produced) at 18 h after induction. Under this condition, the final carotenoid production was only ~140 mg l$^{-1}$ with > 80 mg l$^{-1}$ astaxanthin (70 mg l$^{-1}$

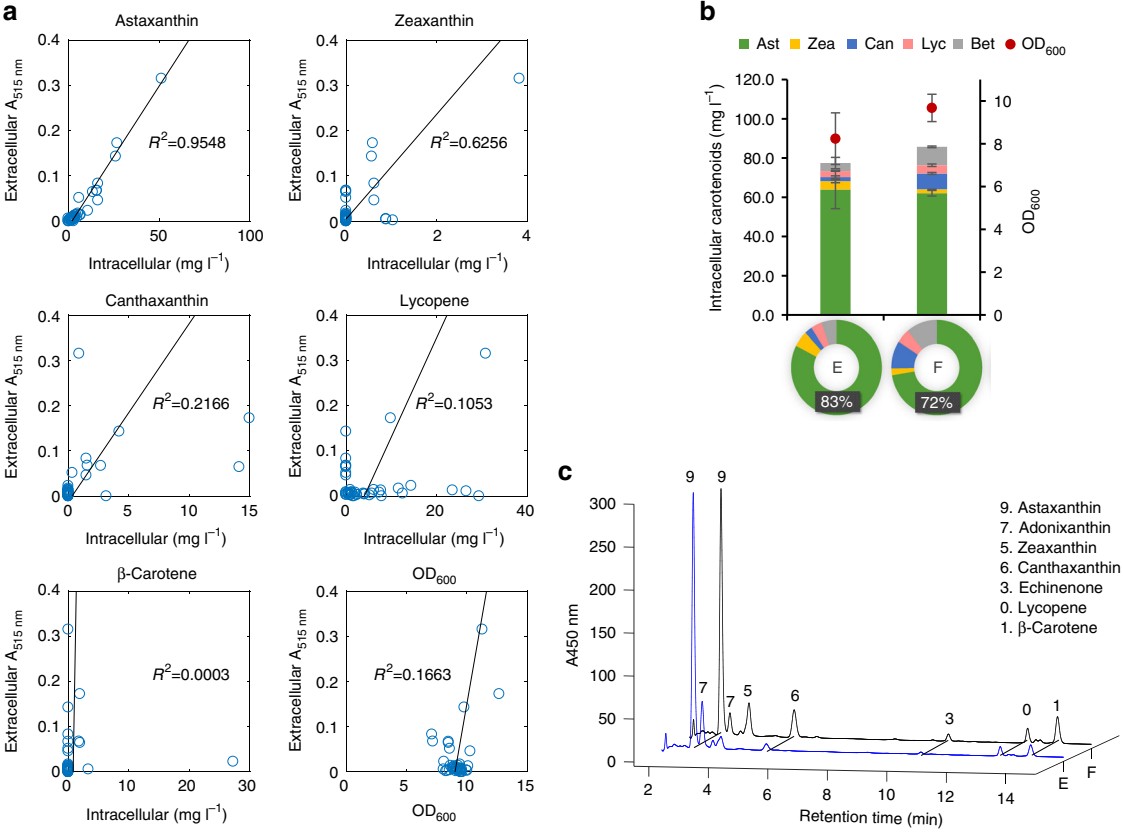

**Fig. 6** Identification of extracellular carotenoids for screening. **a** Correlation between the medium absorbance at 515 nm (A515) and intracellular carotenoid concentration or cell density ($OD_{600}$). Among them, astaxanthin concentration was most strongly correlated with the medium absorbance. **b** The astaxanthin titers and cell density of the two strains, E and F, identified using the medium absorbance as the screening indicator. Error bars, mean ± s.d., $n$ = 3. **c** The HPLC chromatograms of carotenoids produced in the strains E and F

intracellular and 10 mg l$^{-1}$ extracellular astaxanthin, Supplementary Fig. 7b). To mitigate potential intermediates/product toxicity and metabolic burden, we decoupled the growth and production phase by inducing cells at an OD of 70. The late induction resulted in a significant increase in the production of biomass (OD reached 130), total carotenoids (~360 mg l$^{-1}$) and astaxanthin (230 mg l$^{-1}$, Supplementary Fig. 7c).

It is worthy to note that although intracellular carotenoid production increased significantly by decoupling growth and production phase, we did not observe a corresponding increase in the production of extracellular carotenoids (they were in the same range of 15–20 mg l$^{-1}$). This may be due to the limited solubility of these carotenoids in aqueous media[33]. If so, the overall yield and productivity might be improved by simultaneously capturing extracellular astaxanthin in the media with appropriate organics during fermentation, or by simultaneous fermentation and extraction (SFE). To test this hypothesis, we extracted secreted astaxanthin with isopropyl myristate (logP = 7.02, calculated by Alogps[34]) which is both biocompatible and has similar hydrophobicity to that of astaxanthin (logP = 7.4). Indeed, with the introduction of isopropyl myristate, additional 90 mg l$^{-1}$ carotenoids (including 70 mg l$^{-1}$ astaxanthin) were obtained during fermentation, where intracellular carotenoid production was maintained at 370 mg l$^{-1}$ and the aqueous carotenoid titer decreased to 3 mg l$^{-1}$ (Fig. 7). In total, about 320 mg l$^{-1}$ astaxanthin was produced in the bioreactor using SFE, which was ~43% increase in astaxanthin production as compared to that without organic supplementation. Furthermore, the total carotenoids and biomass were increased by 28% (from 360 to 460 mg l$^{-1}$) and 23% (from 130 to 160), respectively.

**MHP for the production of other terpenoids**. To demonstrate the broader utility of MHP, we next examined the production of two colorless compounds, a sesquiterpenoid (nerolidol) and a monoterpenoid (linalool). Nerolidol is a fragrant ingredient widely used in cosmetics (e.g., shampoos and perfumes) and in non-cosmetic products (e.g., detergents and household cleaners). Recent studies indicate nerolidol has many pharmacological and biological activities (antioxidant, anti-microbial, anti-biofilm, anti-parasitic, anti-inflammatory etc)[35]. Linalool is another industrially important fragrant ingredient used in most cosmetic products (body lotions, shampoos, soaps, etc) and has biological activities such as sedative, anxiolytic, and analgesic effects[36]. With our recently identified bifunctional nerolidol/linalool synthase (nls) from *Agrocybe aegerita*, we devised two systems, one expressing nls and ispA (farnesyl pyrophosphate, FPP synthase) from *E. coli* (NA module) to produce nerolidol and the other expressing nls and gpps (geranyl pyrophosphate, GPP synthase) from *Abies grandis*[37] (NG module, Fig. 8a) to produce linalool. We then used MHP to optimize the production of the two molecules. Briefly, three modules were controlled by a combinatorial panel of 3 promoters of different strength (TM1, TM2, and TM3). Simultaneously, nls and ispA (or GPPS) were regulated by another combinatorial panel of 3 RBSs of different translational efficiencies (RBS-4, 6, and 8 in Supplementary Fig. 8). In total, there were $3^5$ = 243 possible combinations. As a proof of concept, we only picked 53 colonies with 1 control strain (Nel1 in Fig. 8b and Lin1 in Fig. 8c). In control strain, all three modules were under the control of strongest TM1 promoter and the genes nls and ispA (or GPPS) were regulated by the strongest RBS (RBS-8). With one round of MHP, we managed to produce 323

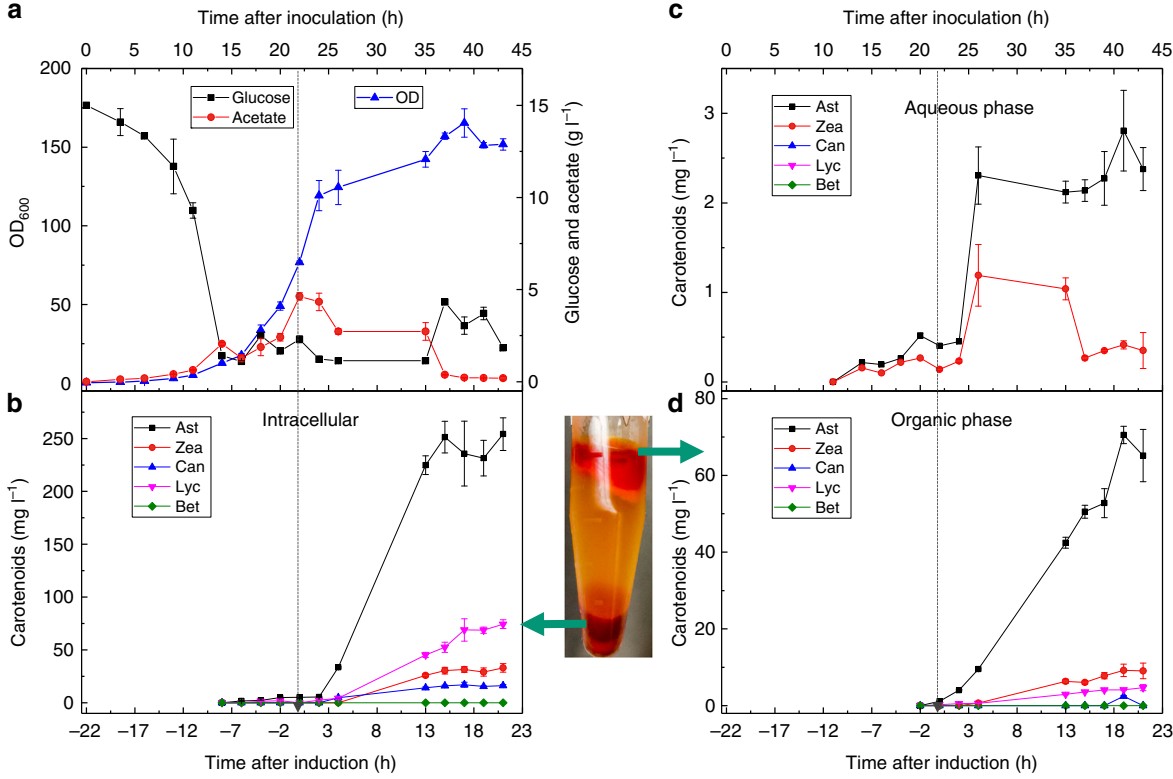

**Fig. 7** Simultaneous fermentation and extraction of astaxanthin. **a** Time-course profiles of OD, glucose and acetate. **b** Time-course profiles of intracellular carotenoid production. **c** Time-course profiles of aqueous carotenoid production. **d** Time-course profiles of organic carotenoid production. The dashed arrow refers to the induction time. Error bars, mean ± s.d., $n = 2$

mg l$^{-1}$ nerolidol (or 43.5 mg l$^{-1}$ OD$_{600}$$^{-1}$) in shake flasks, a 10-fold increase than that of control strain (36 mg l$^{-1}$, Fig. 8b, GCMS spectra shown in Supplementary Fig. 9). Similarly, a 28-fold increase in linalool production was obtained in shake flasks when using MHP, from 2 mg l$^{-1}$ (or 0.2 mg l$^{-1}$ OD$_{600}$$^{-1}$) in the control strain Lin1 to 65 mg l$^{-1}$ (or 8 mg l$^{-1}$ OD$_{600}$$^{-1}$) in strain Lin2 (Fig. 8c; GCMS spectra shown in Supplementary Fig. 9).

## Discussion

In this study, we developed a multidimensional process (MHP) for effectively generating high astaxanthin producers in a 15-step heterologous biosynthetic pathway. In addition, MHP was employed to overproduce two other valuable fragrance molecules, nerolidol and linalool.

Imbalance in pathway fluxes is a major challenge when attempting to improve metabolic pathway performance and limits commercial production of many high-value chemicals by biological systems[38]. The issue becomes ever more complex with the increase in the number of genes/enzymes involved[23,39]. To address this issue, modular pathway optimization represents an efficient solution with distinctive advantages (e.g., modularization of genetic regulatory networks). As genes in the same module are controlled simultaneously, categorizing enzymes with similar turnovers into the same module would be ideal. However, in vivo enzymatic kinetics is difficult to determine. MHP introduces global and local optimization with three-dimensional elements (transcription/ translation/ enzyme). Similar to existing modular approaches, promoters regulate intermodular gene expression enabling the balancing of global metabolic activities. In addition, balancing of intra-modular enzymatic activities involve the use of enzyme variants and regulating translational controls using 5′ UTRs or RBSs. These extended features also distinguish MHP

from existing fractional design-based pathway optimization methods[25,27,40].

Different from other high throughput screening approaches (global transcription machinery engineering or gTME[41], multiplex automated genome engineering or MAGE[42], tunable intergenic regions or TIGR[43]), MHP is more controllable and precise by using pre-calibrated libraries of regulatory elements targeting specific biosynthetic pathways. Coupled with quantitative regulatory elements (promoter or RBS strength), the reverse engineering step allows us to mathematically model the pathway performances which are invaluable for further strain improvements. Mathematical models, if they exist, can guide the optimization of other products or further improvement of existing phenotypes. It is evident that promoter strength of certain modules (e.g., module YZW) or single genes (e.g., crtY) highly correlates to gene transcription levels and metabolite production (Figs. 2g and 3b). However, the effects of RBS variants are less quantitatively correlated to metabolites (Fig. 5c and d), possibly due to yet to be characterized post-translational or regulatory effects. As expected, the twelve strains in Fig. 3c have similar mRNA copies of crtZ or crtW (Supplementary Fig. 10), where the same promoter on the same vector controls these. Attempts were unsuccessful in quantifying the amounts of enzymes produced as the levels of expression were below the detection limit of protein analysis.

Some carotenoids can transverse cell membrane into media[44,45]. If the concentration of extracellular carotenoid correlates with its overall production, measuring the absorbance in the medium may serve as a convenient screening tool. Interestingly, the absorbance of the culture medium was highly correlated with intracellular content of astaxanthin but not the other carotenoids (Fig. 6a). Hence, this has enabled the elimination of the false selection of those colonies accumulating other red

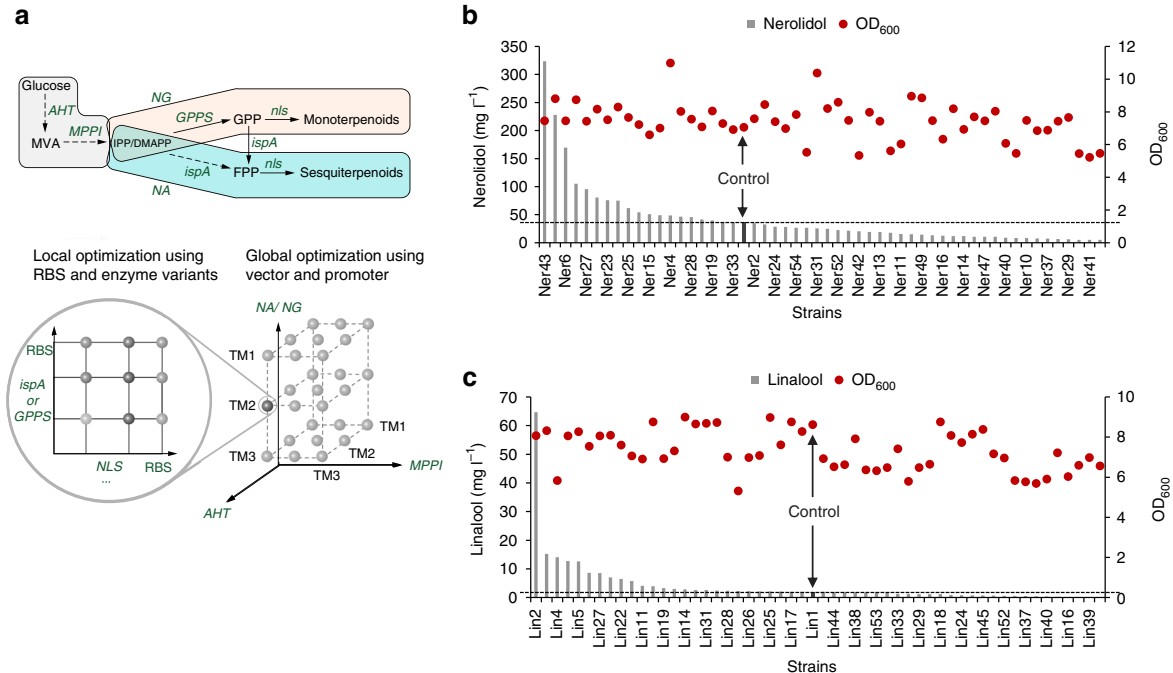

**Fig. 8** MHP for the production of universal compounds. **a** Demonstration of design strategy of MHP. Globally, we used 3 promoters to regulate the transcription of three major modules: the upstream MVA pathway (module AHT), the downstream MVA pathway (module MPPI), the terpenoid pathway (module NG including *nls* and *GPPS* or module NA including *nls* and *ispA*). Locally, we used 3 RBSs to control the relative translation of *nls* and *GPPS* (or *ispA*). In total, there were $3^5$ or 243 combinations. **b** Nerolidol production and cell density for the chosen 54 strains including one control (Ner1, highlighted by an arrow). **c** Linalool production and cell density for the chosen 54 strains including one control (Lin1, highlighted by an arrow). *Nls*, a bi-functional nerolidol/linalool synthase

carotenoids (lycopene or canthaxanthin) instead of astaxanthin. Furthermore, measuring the absorbance of the medium is more objective, thus, making this a better screening assay for the selection of astaxanthin than visual inspection of colonies. With this assay, the screening accuracy has increased from 38% (Fig. 5b) to 100% (Fig. 6b). The high correlation of the absorbance of the media with the production may in part be due to the hydrophobicity of astaxanthin. Consistent with the previous observation[44], the amount of extracellular carotenoids is inversely correlated to their hydrophobicity (Supplementary Fig. 11).

Guided by colorimetric screening of media, we successfully identified two astaxanthin strains (E and F in Fig. 6b and c) of high yields and high purity. A SFE process, was then used to increase the titer (320 mg l$^{-1}$, Fig. 7) and productivity (184 mg l$^{-1}$ day$^{-1}$), simplifying the downstream purification process. To the best of our knowledge, the astaxanthin titer in this study is the highest to date (Supplementary Table 5) and the production rate is 16 times higher than the current algae-based astaxanthin production rate (11.5 mg l$^{-1}$ day$^{-1}$)[46]. In addition, the astaxanthin was found to be 100% pure enantiomer 3 S, 3'S-astaxanthin (Supplementary Fig. 12), distinct from synthetic astaxanthin that comprises a mixture of isomers of (3 S, 3'S), (3 R, 3'S) and (3 R, 3'R).

SFE of lipophilic metabolites, including carotenoids, have previously been reported[44,45], where the overall titers (<10 mg l$^{-1}$) were relatively low. This is due, in part, to the limited cell storage capacity and the overexpression of efflux transporters such as msbA[44,47] and lipid-trafficking proteins (e.g., lptD[47] and the steroidogenic acute regulatory protein[48]) may further increase astaxanthin production.

Extending the study, we produced 324 mg l$^{-1}$ of nerolidol (tenfold higher than the control) and 65 mg l$^{-1}$ of linalool (28-fold higher than the control) in shake flasks in a single round (Fig. 8b and c). When compared to an engineered yeast strain

(95 µg l$^{-1}$)[49], the production of linalool in *E. coli* was more than 680-fold higher. The nerolidol yield in this study is comparable to others (393 mg l$^{-1}$ in shake flasks)[50]. As only about a fifth of the constructs of the whole library was analyzed, a more extensive survey of the rest of the library should identify strains with even better phenotypes. A notable limitation of MHP is that the efficiency of this workflow is constrained by the availability of high-throughput analytical methods for target products. This is mitigated by initially exploring smaller regulatory libraries to speed up the process of finding a satisfactory solution (as described in the MHP workflow in Fig. 4b). In sum, our results demonstrated the uniqueness of MHP serving as a powerful tool with use beyond the scope of this study.

## Methods

**Strain and plasmid construction.** In this study, *E. coli* K12 MG1655 Δ*recA*-Δ*endA*Δ*aroA*Δ*aroB*Δ*aroC* DE3 was used. The genes *hmgS*, *hmgR*, *mevK*, *pmk*, and *pmk* (or *MVD1*) were amplified by PCR using the chromosomal DNA of *Saccharomyces cerevisiae*. The genes *atoB* and *idi* were amplified from *E. coli* MG1655 genomic DNA. All the genes were cloned into two plasmids, p15A-spec-hmgS-atoB-hmgR (L2-8) and p15A-cam-mevK-pmk-pmd-idi (L2-5). The genes in the lycopene biosynthetic pathway (*crtEBI*) amplified from the pAC-LYC plasmid[51] was introduced into p15A-kan-crtEBI-ispA plasmid. Two *crtY* genes (lycopene cyclase), one from *Pantoea ananatis* LMG20103 and the other from *Uncultured marine bacterium* HF10_19P19, were codon-optimized and cloned into the plasmid pET11a or p15A-amp, forming β-carotene plasmids pET11a-*crtY* or p15A-amp-*crtY*. Subsequently, the two codon-optimized *crtZ* genes (*crtZ1* from *Sulfolobus solfataricus* P2 and *crtZ2* from *Pantoea ananatis* LMG20103) and *crtW* genes (*crtW1* from *Anabaena variabilis* ATCC 29413 and *crtW2* from *Brevundimonas sp.* SD212) were inserted into the β-carotene plasmids with four combinations, resulting in astaxanthin plasmids pET11a-*crtYZW* and p15A-*amp-crtYZW*. The nerolidol/linalool synthase (*nls*) gene were from the *Agrocybe aegerita* and the *GPPS* gene from *Abies grandis* were synthesized by IDT technology and cloned into p15A-kan vector. The p15A series of plasmids were derived from pAC-LYC plasmid[51] by mutating the p15 origin to make the plasmids compatible to each other[52]. Plasmid compatibility was checked by comparison of the copy number of each plasmid after 10 generations. The strains and plasmids used in this study were summarized in Supplementary Table 1.

**Construction of promoter library**. To mutate T7 promoter and construct the promoter library, standardized modular DNA fragments was designed for constructing multi-gene plasmids by the refined Cross-Lapping In Vitro Assembly (CLIVA) method[53]. The refined CLIVA method is more robust and simpler than most of existing cloning methods, especially for multi-piece assembly (up to 10 pieces) with very high accuracy and thus ideal for the construction of combinatorial library. Primers used for the construction of T7 promoter variant library were degenerate oligos I-T7P_lib-F and I-T7P_lib-R (Supplementary Table 6). Specifically, the plasmids with the promoter library generated were subsequently transformed into E. coli cells and the resulting cells were plated on LB-agar plates with appropriate antibiotics. One hundred colonies were picked and grew in modified 2XPY medium. The fluorescent signal of expressed eGFP (excitation at 488 nm and emission measurement at 509 nm) of each colony was correlated with their promoter sequence accordingly. After characterization, the three T7 promoter variants, TM1, TM2 and TM3 were chosen. The relative transcriptional efficiencies of three T7 promoter variants together with native T7 promoter were characterized by fluorescence signal of eGFP (Supplementary Fig. 1a and Supplementary Table 7). For the rest of the study, these four promoters were used instead of the whole promoter library.

**Construction of RBS library**. Similar to promoter library, two RBS libraries were designed for crtZ and crtW using the degenerate primer I-RBSlib-crtZ2-F and I-RBSlib-crtZ2-R, I-RBSlib-crtW2-F and I-RBSlib-crtW2-R (Supplementary Table 6), respectively. RBS design was aided by RBS Calculator (version 2.0, https://salislab.net/software/)[54]. The predicted translation efficiencies of the RBS libraries for crtZ and crtW were listed in Supplementary Table 3 and 4. Different from promoter experiment, the RBS efficiency was not experimentally calibrated but calculated by RBS Calculator. Instead, a plasmid mixture with different RBSs was generated by extracting plasmid from a pooled collection of colonies on the plate. Subsequently, the plasmid mixture was transformed into the cells. Based on color intensity, dozens of colonies were picked with red colors and one or two with low production as control. The same procedure was applied for the generation of RBS combination for nls and GPPS (or ispA) but with pre-calibrated RBSs (Supplementary Fig. 8). The fluorescence was quantified using the MACSQuant® Analyzer 10. A violet laser (405 nm) and a 450 ± 50 nm emission filter were used to measure CFP fluorescence. For all samples, 100,000 cells were recorded in each flow cytometry experiment.

**Construction of dual library of promoter and RBS**. First, a plasmid library was obtained with different promoters (first round). Following that, the first round plasmid library was used as template to mutate the RBSs of nls and ispA or GPPS with a pre-calibrated RBS library (RBS-4, 6 and 8 in Supplementary Fig. 8), resulting in a second-round plasmid library. The second-round plasmid library including both different promoters and RBSs was used for the following MHP screening experiments.

**Batch medium**. The batch medium for fermentation contained 15 g l$^{-1}$ glucose, 2 g l$^{-1}$ (NH$_4$)$_2$SO$_4$, 4.2 g l$^{-1}$ KH$_2$PO$_4$, 11.24 g l$^{-1}$ K$_2$HPO$_4$, 1.7 g l$^{-1}$ citric acid, 0.5 g l$^{-1}$ MgSO$_4$ and 10 ml/l trace element solution, pH 7.0. The trace element solution (100×) contained 0.25 g l$^{-1}$ CoCl$_2$·6H$_2$O, 1.5 g l$^{-1}$ MnSO$_4$·4H$_2$O, 0.15 g l$^{-1}$ CuSO$_4$·2H$_2$O, 0.3 g l$^{-1}$ H$_3$BO$_3$, 0.25 g l$^{-1}$ Na$_2$MoO4·2H$_2$O, 0.8 g l$^{-1}$ Zn(CH$_3$COO)$_2$, 5 g l$^{-1}$ Fe(III) citrate and 0.84 g l$^{-1}$ EDTA, pH 8.0.

**Feed medium**. The feed medium for bioreactor contained 500 g l$^{-1}$ glucose and 5 g l$^{-1}$ MgSO$_4$.

**Seed medium**. Seed medium for pre-culture and shake flask experiments is the same as the batch fermentation medium.

**Modified 2XPY medium**. Modified 2XPY medium contained 20 g l$^{-1}$ peptone, 10 g l$^{-1}$ yeast extract, 10 g l$^{-1}$ NaCl, 30 mM HEPES, 10 g l$^{-1}$ glycerol, and 0.5% (v v$^{-1}$) Tween 80, pH 7.0[47].

**Tube and shake-flask method**. Cell cultures in tubes were initially grown at 37 °C with the shaking speed of 250 rpm and were induced by 0.03–0.1 mM IPTG (or 1~20 mM lactose) when OD$_{600}$ reached 1.0. For simultaneous fermentation and extraction (SFE) of astaxanthin, 20% (v/v) of isopropyl myristate was supplemented on top of the cell culture. After induction, cells were further incubated at 30 °C for 24 h before harvest. The media were supplemented with appropriate antibiotics (100 µg ml$^{-1}$ ampicillin, 34 µg ml$^{-1}$ chloramphenicol, 50 µg ml$^{-1}$ kanamycin, and 50 µg ml$^{-1}$ spectinomycin) to maintain corresponding plasmids.

**Cultivation method of linalool and nerolidol strains**. The linalool or nerolidol strains were cultured in the modified 2XPY medium for 2 days at 30 °C and with the shaking speed of 250 rpm. Specifically, fresh cells (20 µl) were inoculated into 1 ml fresh media in 14 ml BD Falcon™ tube and covered with 200 µl dodecane (with 100 mg l$^{-1}$ β-caryophyllene as internal control).

**Batch process in bioreactor**. Two 250 ml Mini Bioreactors (Applikon Biotechnology) were used in this study. To prepare the seed culture, a single colony from agar plate was inoculated into 1 ml modified 2XPY medium at 37 °C overnight and the cells were then transferred into 10 ml seed medium for 10 h. Two milliliter of the grown cell suspension (OD of ~5) was inoculated into 100 ml batch medium in the bioreactor. During the fermentation, dissolved oxygen levels were maintained at 30% (800–2000 r.p.m.) by supplying filtered air at a gas rate of 1.5 vvm. The pH of the culture was controlled at 7.0 using 28% ammonia solution. The cell culture in the bioreactors was grown at 37 °C for 9 h (OD reached ~4) and then was induced by 0.1 mM IPTG. The culture was maintained at 30 °C for another 20 h.

**Fed-batch process in bioreactor**. Different from batch process, once OD reached about 6–8, feed medium was fed into the bioreactor at the rate of 0.9–1.8 g h$^{-1}$ for about 4 h and stopped until DO increases abruptly (or glucose concentration in the broth decreased below 5 g l$^{-1}$ by offline measurement). The feeding cycle was repeated until the end of the fermentation process. For early induce fed batch process, cells were induced by 0.1 mM IPTG when OD reached about 15 (~12 h from inoculation). For late induce fed batch process, cells were induced when OD reached 60–70 (20–22 h from inoculation). For two phase fed-batch process, cells were induced when OD reached 60–70 (~22 h from inoculation). After induction, 20% (v/v) of isopropyl myristate was supplemented into the bioreactor.

**Carotenoid extraction**. Intracellular carotenoids were extracted from cellular pellets according to the acetone extraction method[51]. Briefly, 10–50 µl bacterial culture was collected and centrifuged to remove the supernatant. Cell pellets were washed once with PBS and were resuspended in 20 µl of water, followed by addition of 180 µl of acetone. The resulting solution was incubated at 50 °C for 5 min with shaking speed of 1000 rpm. Extracellular carotenoids were extracted using ethyl acetate. One milliliter of bacterial culture was centrifuged for 5 min at 20,000 × g, and 800 µl of clear supernatant was transferred into a new 1.5 ml Eppendorf tube with 200 µl of ethyl acetate. The resulting solution was subsequently well mixed in dark room at 4 °C and centrifuged for 10 min at 20,000 × g. After centrifugation, 50 µl of ethyl acetate with extracted carotenoids (the top layer) was used for HPLC analysis.

**Carotenoid quantification by HPLC**. For strains that only produced lycopene, the lycopene content was quantified by absorbance at 472 nm using Synergy™ HT Multi-Detection Microplate Reader and calculated by interpolating the standard curve. For strains that produced mixture of carotenoids, HPLC analysis was performed similarly as described previously[9]. Briefly, the analysis employed an Agilent 1260 Infinity LC System equipped with a ZORBAX, Eclipse Plus C18, 4.6 × 150 mm$^2$, 5 µm column. The mobile phase and gradient used were as follows. Initially, the analysis used isocratic condition (88% methanol, 2% solvent water and 10% ethyl acetate) for 5.5 min at a flow rate of 1 ml min$^{-1}$. Ethyl acetate was gradually increased from 10% to 48% (at the same time, methanol was decreased from 88% to 50% over 5.5 min at 1.5 ml min$^{-1}$). Finally, another isocratic condition (50% methanol, 2% solvent water and 50% ethyl acetate) was maintained at 1 ml min$^{-1}$ for 4 min. The whole process finished at 16 min and all the carotenoids were detected at 450 nm. Standard curves were generated using commercial standards: lycopene, β-carotene, astaxanthin, canthaxanthin (Sigma-Aldrich, St. Luis, MO, USA), and zeaxanthin (Santa Cruz Biotechnology, Dallas, TX, USA). Cell mass was calculated by correlating dry cell weight with OD$_{600}$ for use in calculating carotenoid contents (µg carotenoids per gram DCW, dry cell weight). The chirality of our astaxanthin was analyzed by the chiral column, Chiralcel OD-RH, 150 × 4.6 mm$^2$ (Daicel Corporation, Osaka, Japan), 5 µm particle size. The mobile phase was 85% hexane and 15% acetone, at 1 ml min$^{-1}$ flow rate.

**Carotenoid identification and quantification**. The carotenoids, especially those without chemical standards (e.g., adonixanthin and echinenone), were analyzed by Agilent 1290 Infinity II UHPLC System coupled with 6230B TOF MS platform. One microliter of purified carotenoids in acetone was injected into the Agilent ZORBAX Eclipse Plus C18 column. Separation was carried out at a flow rate of 0.4 ml min$^{-1}$. The mobile phase and gradient used were as follows. The analysis started from 10% water, 50% methanol, and 40% acetonitrile. Methanol was gradually decreased from 50% to 10% in 1 min, with the acetonitrile increased from 40% to 80%. This condition (10% water, 10% methanol, and 80% acetonitrile) was maintained for 1.1 min, followed by methanol increased to 100% within 0.1 min. The condition (100% methanol) was continued for 7.7 min. The whole analysis finished at 9.3 min. Mass spectrometry was operated to scan 100–1000 m z$^{-1}$ in ESI-positive mode with 4000 V capillary voltage. Nebulizer gas was supplied at 35 psig and dry gas flow was 10 l min$^{-1}$. Gas temperature was set at 325 °C. Shealth gas was set at 350 °C and 12 l min$^{-1}$. Retention time was determined with chemical standards or calculated based on chromatography profile for those carotenoids without standards. Carotenoid concentrations were calculated based on the peak area of each compound extracted by their corresponding m/z value (lycopene, 536.438; β-carotene, 536.438; β-cryptoxanthin, 552.433; echinenone, 550.417; 3'-hydroxyechinenone, 566.412; zeaxanthin, 568.428; canthaxanthin, 564.397; adonixanthin, 582.407; adonirubin, 580.392; astaxanthin, 596.387).

**Quantification of terpenoids**. The terpenoid samples were prepared by diluting 10–50 μl of organic layer into 1000 μl hexane. The samples were analyzed on an Agilent 7890B gas chromatography equipped with an Agilent 7200 Accurate-Mass Quadrupole Time-of-Flight (GC/MS). Samples were injected into Agilent VF-WAXms column with a split ratio of 10:1 at 240 °C. The oven program started at 100 °C for 2 min, then the temperature was raised up at 10 °C min$^{-1}$ until 240 °C and maintained at 240 °C for another 2 min. The compound concentrations were calculated by interpolating with a standard curve prepared by authentic standards. Mass spectrometer was operated in EI mode with full scan analysis ($m/z$ 30–300, 2 spetra s$^{-1}$).

**Quantification of extracellular metabolites**. Cell suspension (500 μl) were centrifuged for at $16,000 \times g$ for 10 min. The supernatant was further filtered by 0.2 μm membrane. For HPLC analysis, 10 μl of the filtered supernatant was used to quantify the concentrations of metabolites, including glucose and acetate, with the Agilent 1260 Infinity LC System. The mobile phase (5 mM sulfuric acid) was used to flow through an Aminex column (Bio-Rad HPX-87H column) at a rate of 0.7 ml min$^{-1}$.

**RNA purification and quantitative real-time PCR**. Total RNA from *E. coli* cells was prepared using PureLink® RNA Mini Kit (Thermo Fisher Scientific, Waltham, Massachusetts, USA) according to the manufacturer's instructions. All the samples were done in triplicates. The integrities of RNA samples were checked by formaldehyde agarose gel. RNA samples (800–1000 ng) were reverse transcribed in a total volume of 40 μl solution at 37 °C for 60 min and the reaction was subsequently terminated at 95 °C for 5 min. The cDNA concentrations were then determined using QuantStudio 3 Real-Time PCR System (Thermo Fisher Scientific) with SYBR green I detection. Briefly, a reaction mixture (25 μl total volume) contains 1 × SYBR green PCR buffer, 200 nM primer mix, 2.5 mM MgCl2, 0.75 U of iTaq DNA polymerase (i-DNA Biotechnology Pte Ltd, Singapore). Real time PCR was performed with an initial denaturation of 3 min at 95 °C, followed by 40 cycles of amplification (20 s at 95 °C, 20 s at 55 °C, and 20 s at 72 °C). Post-PCR melting curve analysis was performed in the range of 60 °C to 95 °C at 0.15 °C s$^{-1}$ increment. The primers used for real time PCR were in Supplementary Table 6, and the reference gene used for normalization of real time PCR data was cysG[20].

**Data availability**. All data supporting the findings of this study are available in the article, Supplementary Information, or upon request from the corresponding author.

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

## Acknowledgements

We acknowledge Dr Nicholas David Lindley, director of Biotransformation Innovation Platform and A/Prof Zhou Kang (National University of Singapore) for advice and insightful discussion. This research was supported by Singapore-MIT Alliance for Research and Technology (SMART) and Biotransformation Innovation Platform, Agency for Science, Technology and Research (A*STAR), Singapore.

## Author contributions

C.Z. and H.P.T. conceived the project. C.Z. designed the experiments and analyzed the data. C.Z., V.Y.S., and X.C. did the experiments. C.Z. and H.P.T. wrote the manuscript. All authors contributed to the discussion and approved the final manuscript.

## Additional information

**Competing interests:** The authors declare no competing interests.

