## [Peer Review File · Nature Communications]

Reviewers' comments:

Reviewer #1 (Remarks to the Author):

The authors present an interesting study of a multidimensional optimization by changing promoter strength, RBS site, and gene. Such a multidimensional optimization does improve strains (yet, this work looks quite similar to work done by the Zhou group at Illinois in yeast using the COMPACTER approach). The combinatorial space of promoters and RBS sites has certainly been explored by others including the work of Salis group at Penn State and the Voigt group at MIT. These overlaps should be more explicitly stated in this manuscript. In this regard, the novelty here is a bit overstated. That said, there is promise in the work. However, several aspects need to be improved.

A major premise of this work is the screening approach. Specifically, the underlying assumption is that extracellular concentration is proportional to intracellular production. However, this is not fully proven. In particular, while the authors show that there is different excretion for different carotenoids, it is not seen whether these values are constant and not a function of intracellular accumulation. This should be done at least in hindsight to prove the premise (based on the strains identified and observed). The issue of false negative and positive rates are not known and should be at least quantified to some extent.

The meaning of error bars are not shown in the legends. Also, the number of replicates is not shown. This is not consistent with the quality control standards of Nature Communications.

In the abstract, the value of 320 mg/L is a titer, not a yield.

The authors use the term “robust” to describe their MHP in most of the text, except for the title. The title is likely most correct. I am not seeing why / how this process is robust. This is a random sampling approach with a built-in assumption of a hill climbing trajectory—which is not a robust algorithm for global optimization in many cases. This should be corrected.

Reviewer #2 (Remarks to the Author):

The review is copied below. For ease of access to comments a Word document is also attached (containing identical information).

In the current manuscript, Zhang et al have developed a method they call ‘Multidimensional Heuristic Process’ for pathway optimisation. Before assessing this manuscript is necessary to provide background on multivariate approaches for combinatorial metabolic engineering. The

first approach described in this family of approaches is multivariate modular metabolic engineering (MMME)¹. In this approach, enzymes are organized into operon modules comprising a set of pathway steps, and expression of each operon is varied simultaneously. Generally speaking, expression levels have been titrated in several ways, as noted in the current manuscript: gene copy number, promoter expression level, and ribosome binding site. Using this approach, an optimal flux balance between modules is identified. This optimality may arise from avoiding accumulation of intermediates which might be toxic and/or generally balancing flux through the pathway overall (it may be unclear why the particular expression balance is optimal). The multiplexed approach means that a very large sample space can be sampled as long as there is a facile and high-throughput screening technique. The EDASPO method developed out of the same lab² introduced a predictive element to the multivariate approach, allowing a much smaller solution space to be sampled to provide sufficient data resolution allowing prediction of module titration requirements that would allow further titre improvements.

In this work, the authors expand the approach to modulating both expression level (through screening libraries of regulatory elements, which includes promoters driving multigene operons as well as 5'UTRs to modulate expression of individual genes in each operon) combined with swapping specific enzyme variants from either natural sources or engineering products (so that specific activities/selectivities are also screened). These approaches (1) expanded the dimensions (and consequently the resolution and size of the solution space), improving the overall utility of the multivariate approach, and (2) allowed them to deal with a particular problem in the astaxanthin pathway, where two enzymes (crtZ and crtW) catalyse multiple different reactions each, and an appropriate balance of these reactions is required in order to optimise (maximise) flux through to the desired end product. These novel approaches and their impact are surprisingly not at all elucidated in the abstract, leaving the reader unsure of the significance of the study and indeed of the details of the method itself.

The authors also developed a high-throughput assay allowing them to differentiate between the target compound, astaxanthin, and other red compounds (including lycopene, canthaxanthin). This was based on the observation that astaxanthin can traverse membranes far more readily than the carotenoid compounds, resulting in accumulation in the extracellular medium – providing a facile method to screen spectrophotometrically at the absorption maximum for astaxanthin. The best strain from the combinatorial engineering and screening strategy was then fermented in batch process bioreactor, with some process optimisation to improve growth rates, yielding impressive titres of 320 mg/L astaxanthin out of a total of 460 mg/L total carotenoids produced, with productivities exceeding the current industrial process. Finally, the broader utility of the MHP approach (using promoter and RBS combinations for three modules) was tested on two other isoprenoids, one sesquiterpene and one monoterpene. Good results were obtained for the monoterpene product, but less impressive for the sesquiterpene (the lack of high throughput selection method limited execution of this component of the project).

The manuscript suffers from numerous communication problems, in particular around style, use of language and grammar, and structure. This makes it a bit impenetrable in places, and the overall significance of the story might easily be lost as the context and background is not clearly explained in the introduction. Notwithstanding this, the manuscript is an original contribution and demonstrates a remarkably effective approach at increasing production levels for the target

compound, astaxanthin. This is likely to be generalizable, in particular for target compounds where a facile high-throughput assay is available: less compelling results are presented for the other isoprenoid based target compounds, but the method is far less robustly applied in those cases and no HTP screening method exists, significantly curtailing the ability to effectively identify high producers. The study is ambitious in theoretical scale (although less so in execution). Collection of a bit more data (as outlined below) would make the paper more robust and would likely increase the scope and impact of the findings – specifically, the authors are using titrations of transcription, translation, and enzyme activity – but they are measuring none of these directly – they only measure the metabolites. The only expression level data presented is reporter gene activity, which doesn't necessarily scale directly, and they use theoretical expression strengths also in places. Unsurprisingly, correlations are not identified between these data and the metabolite accumulation data. For completion, it would be appropriate to pay more attention to quantitative measurement at all stages of expression to allow meaningful correlative analysis with metabolite data.

It is likely that this multi-dimensional approach of titrating at various levels of expression and enzyme activity will find broader utility in metabolic engineering for different products, especially where high throughput analysis for the target product is available.

General

1. English use and grammar needs to be improved throughout the text
2. Communication could be significantly improved by working on the text in the introduction. Such improvement would place the results section in a much better context. A proper description of the original multivariate approach as well as all of the subsequent improvements and modifications to the overall theme should be done in the introduction as part of the literature review
3. A number of breaches of scientific writing protocol are made (e.g., presenting data from the results as part of the introduction)
4. It's a pity that strains from the library were discarded in favour of focussing strongly on astaxanthin producing strains (page 7). A far more detailed picture of the expression balance requirements for different intermediates and end-products might have been obtained by characterising and examining a larger range of strains.
5. A lot of assumptions are made about relationships between translation rates and enzyme activities (e.g. Fig 4c and other statements in the text); however, the actual transcription/translation is never measured – assumptions are made based on theoretical transcription levels and reporter gene data. This is problematic for the current study, since it is describing and attempting to parameterise the approach. The manuscript would be improved by measuring transcription and ideally also translation quantitatively. This should be feasible in a subset of useful strains, as the refined sets are relatively small. Such analyses might reveal substantially more about the driving factors behind the observed phenomena.
6. No growth data are presented in the results. This is a significant oversight, since pathway imbalances commonly cause growth defects. A difference in growth rate might significantly affect the observed carotenoid accumulation/titre as a given time point. On the other hand, this can be standardised for by presenting ppm as is done frequently. Notwithstanding this, growth profiles can be quickly assessed by microtitre growth plate readers. This should be done as the growth profile is a standard part of any metabolic engineering strategy and may shed more light

on the physiology and phenotypes. Indeed, the authors only briefly mention growth defects as a result of induction of the pathway genes in the bioprocess section. This confirms the importance of gathering more growth data throughout this work.

7. In the results I find it a bit painful to move between ppm and titre. In general, titre is the more commonly-reported result
8. Statistical support for some of the comments made is not provided
9. A limitation in the execution of this protocol is the need for a high throughput screen to effectively exploit the approach. This is evident in the relative success for the different products targeted. This limitation should be properly explained and explored in the discussion, and for completeness, it should be referred to in the abstract.

Abstract

10. From the abstract, I have absolutely no idea what the ‘multidimensional heuristic process’ is – despite it being the title of the paper. A description of what the process is should be embedded in the abstract since it’s the central focus of the paper. The product itself is almost incidental. One should be able to determine from the abstract what the novelty of the story is, and it’s impossible to do that from the current abstract.
11. It’s always a bit suspicious when titres are reported for one compound but only fold improvement for others (astaxanthin vs. nerolidol and linalool)

Introduction

12. From a biotechnology perspective, it should probably be noted that astaxanthin is approved as an oral supplement (despite having very low oral absorption) and as a food colourant and feed additive.
13. Lines 53-56: the logical gap here is that the previously engineered strains presumably accumulate a large amount of beta-carotene as a result of upstream pathway engineering, but that this is not converted efficiently into astaxanthin (despite trying several different ketolases and hydroxylases). There is context missing in this section, as the authors focus in this paragraph on (a) sufficient production of lycopene, or (b) problems with crtY, crtW, and crtZ. There is plenty of information available in the literature on accumulation of products that are intermediates in this pathway (primarily lycopene and beta-carotene, for which reasonably high titres can be achieved).
14. Line 64: when multivariate modular metabolic engineering is first mentioned, it would be useful to explain what it is and what its utility is before pointing out the restriction of the method and then introducing the modified EDASPO method
15. The EDASPO method, and in particular the differences of this method compared to the MMME method, should be described (not just the advantages, but the methodological improvement); and the acronym should be defined at first use

16. It is unclear from the context of the text whether the data from the EDASPO method for lycopene and beta-carotene presented on line 68 was published previously (in which case it should be cited both here in the text and in the figure caption, and I do not see a need to re-publish the data) or whether it was generated for this work (in which case it should be in the Results).

17. Similarly the data in the results should not be referred to in the Introduction.

18. Lines 74-75: 'In dealing with complex pathways, previous studies have focused on using either one or two of the controlling dimensions (gene copy 24, 26, promoter 24, 25, 27, 28 or ribosome binding site (RBS) 29, 30), which leaves either local or global pathway not sufficiently optimized and hence limiting the final yields'. It is unclear what the authors mean by this and why sufficient optimisation is not obtained using these approaches. Thus, the overall significance of the work is unclear.

Results

19. Lycopene was used as a screen in several places. It is worth keeping consideration of the fact that lycopene is not always a direct readout of pathway flux, since lycopene is highly sensitive to oxidation and cellular stress results in production of reactive oxygen species. This can affect lycopene titres as the oxidised products are not red³

20. It is unclear what the arrows on Figure 2b mean from the caption. You have to guess from the text on line 104

21. Line 97: it's unclear why Fig 3a is referred to here

22. Line 105: the authors state that strain 22 had the highest astaxanthin, but this is incorrect based on figure 2b. Strain 2'2 clearly has much higher levels of astaxanthin (though decreased overall flux to the carotenoid family, and restricted product profile – it has 93 % astaxanthin and only a small amount of lycopene. This strain therefore is very interesting since it has the potential to produce a high level of specific product, and I am confused as to why this strain was not further investigated.

23. Line 108-110: 'Consistent with the hypothesis, the relative expressions of ketolase and hydroxylase have significant effects on the production of astaxanthin and total carotenoids (Fig. 2b).' The statement does not align with the cross-reference to the figure, which only shows the carotenoid accumulation data and not the relative expression levels (which I think are deduced from the GFP expression data shown in suppl. figure 1a, though that is not clear from the text, and this is a poor way to do the assessment – expression levels can be easily measured using RT-PCR and compared with the product levels in the relevant strains to determine if a correlation exists. And in this case, a proper statistical analysis should be done.

24. Please convert ppm to titre for key strains (both can be included).

25. The sample size for the sesquiterpene and monoterpene study was small compared to the theoretical sample size, but this is understandable considering that there is no high throughput screen for these products

Discussion

26. The fold increases for the nerolidol/linalool work are not bad, and the linalool titres are very good. But the final titres of nerolidol are not impressive compared to the best strains currently published (and in general for a sesquiterpene). It would be polite to acknowledge that,

and note the studies which have produced high titres of these compounds to date and why they have been more successful (this might suggest a next step engineering approach that the authors could include in future studies applying the MHP for this target compound. Similarly, an analysis of the engineering steps required to improve monoterpene titre might shed insights on future steps for that target)

27. Lines 284-288: carotenoids have been produced at least an order of magnitude higher than the levels reported here for astaxanthin. It will probably be a while before astaxanthin cell titres reach levels that require engineering of export components. Furthermore, as noted, this is one carotenoid which is reasonably effectively exported from the cell. So I cannot see that engineering transporters are the next step. Instead, identification of other bottlenecks for conversion through the pathway would be required.

Materials and methods

28. The cultivation method for terpenoid production is missing. Noteworthy is the fact that these C10 and C15 terpenoids are highly volatile and cultures are typically performed in sealed flasks using an organic phase to sequester the terpenoids to minimise loss. Since an organic phase sample was used for the analytics, I presume that the two-phase cultivation was performed, but the details for this component must be described.

29. It is appropriate to highlight that better in the discussion that the method is constrained somewhat where there is a lack of high throughput selection method, as this is a limitation of the approach.

References cited

1. Ajikumar, P. K.; Xiao, W.-H.; Tyo, K. E. J.; Wang, Y.; Simeon, F.; Leonard, E.; Mucha, O.; Phon, T. H.; Pfeifer, B.; Stephanopoulos, G., Isoprenoid pathway optimization for taxol precursor overproduction in *Escherichia coli*. *Science* 2010, 330 (6000), 70-74.
2. Zhang, C.; Zou, R.; Chen, X.; Stephanopoulos, G.; Too, H. P., Experimental design-aided systematic pathway optimization of glucose uptake and deoxyxylulose phosphate pathway for improved amorphadiene production. *Appl Microbiol Biotechnol* 2015, 99 (9), 3825-37.
3. Bongers, M.; Chrysanthopoulos, P. K.; Behrendorff, J. B. Y. H.; Hodson, M. P.; Vickers, C. E.; Nielsen, L. K., Systems analysis of methylerythritol-phosphate pathway flux in *E. coli*: insights into the role of oxidative stress and the validity of lycopene as an isoprenoid reporter metabolite. *Microb Cell Fact* 2015, 14 (1), 1-16.

Reviewer #3 (Remarks to the Author):

This manuscript describes the improvement of natural products biosynthesis in *E. coli* through a so-called multidimensional heuristic process (MHP), which combines optimization of promoters, RBS and enzyme source for pathway enzymes. Although high productivity and yield of natural products such as astaxanthin was achieved using this process, the novelty and technical advance of the work seem to be insufficient to meet the publication criteria of *Nature Communications*. The difference between the reported method and the previously reported modular pathway optimization approaches is not big enough to distinguish it as an inspiringly new method. Some more issues are listed below:

1. The manuscript is not well structured and the findings are insufficiently discussed. In particular, the technical details are not well presented.
2. Although the authors emphasize astaxanthin synthetic pathway as challenging in pathway optimization since it is a long pathway involving 15 reactions, the experiments described in this paper are merely combination of RBS and enzyme sources (two for each enzyme) for CrtW and CrtZ, whereas the more labor-intensive promoter optimization for multiple pathway enzymes carried out using the previously reported EDASPO method was not described in details. Especially, for the abbreviation EDASPO, even the full name was not mentioned. In addition, the failure of EDASPO method in production of high astaxanthin was attributed to its inability to deal with the complex biosynthetic pathway, without further explanation and discussion on the possible synergy among different dimensions of regulatory factors.
3. The two-phase fermentation process developed for improved astaxanthin production is inspiring for carotenoids biosynthesis, but the optimization of the organic solvent layer may be introduced in more details, and more discussion may be included on extracellular secretion of carotenoids and two-phase fermentation of lipophilic products. Literatures such as “Appl Microbiol Biotechnol (2016) 100:869–877” and “Yeast 2010; 27: 983–998” may be referred to.
4. For astaxanthin biosynthesis in engineered microorganisms, a recent publication on *Kluyveromyces marxianus* might be included (Lin Y J, Chang J J, Lin H Y, et al. Metabolic Engineering a Yeast to Produce Astaxanthin. Bioresource Technology, 2017. doi.org/10.1016/j.biortech.2017.07.116)

Responses to Reviewers:

We wish to thank the reviewers for their insightful comments and invaluable suggestions. We have revised the manuscript and have addressed all queries in detail. The point-by-point response to the reviewers' comments are included below.

List of actions (main changes in the revised manuscript and the revised parts are highlighted with blue color)

- 1. We have engaged a senior native English speaking scientist/administrator to help improve the command of language and matters of communication.*
- 2. We have revised the Abstract section to better capture the essence and novelty of this study.*
- 3. The introduction has been re-structured to provide the background information on multivariate modular metabolic engineering (MMME) method and experimental design-aided systematic pathway optimization (EDASPO) method. More importantly, the definition, methodological improvements and differences between this study and other existing metabolic pathway engineering approaches are described in detail in the revised version of this manuscript.*
- 4. All the data relating to the results in the Introduction section have moved to the Results 3.1 section. Fig. 2 in the main text has replaced the previous supplementary Fig. 1 and 2.*
- 5. New data is included to support the use of extracellular carotenoids (mainly astaxanthin) as a screening tool. New supporting results are now included on the relationship between intracellular and extracellular carotenoids, and the correlations between medium absorption and different extracellular carotenoids. In addition, the transmembrane diffusional capability of carotenoids and the relationships to hydrophobicity are included (Fig. 6a, Supplementary Fig. 5, 6 and 11).*
- 6. New transcriptional analyses data using real-time quantitative PCR are included for monitoring the expressions of crtY, crtZ and crtW (Supplementary Fig. 1 and 11).*
- 7. False positive results of colony-color based screening are included in Fig. 5.*
- 8. For consistency, all the data previously expressed as 'ppm' intracellular content are now in 'mg/L' titers. Similarly, biomass data (OD600) are included in the relevant figures.*
- 9. The discussion section now includes an emphasis on the uniqueness of this study as compared to existing methods, the transmembrane diffusional capability of carotenoids in the development of screening methods, simultaneous fermentation and extraction process. Issues pertaining to the limitation of this study are also included.*

Reviewers' comments:

Reviewer #1 (Remarks to the Author):

The authors present an interesting study of a multidimensional optimization by changing promoter strength, RBS site, and gene. Such a multidimensional optimization does improve strains (yet, this work looks quite similar to work done by the Zhou group at Illinois in yeast using the COMPACTER approach). The combinatorial space of promoters and RBS sites has certainly been explored by others including the work of Salis group at Penn State and the Voigt group at MIT. These overlaps should be more explicitly stated in this manuscript. In this regard, the novelty here is a bit overstated. That said, there is promise in the work. However, several aspects need to be improved.

A few useful stand-alone methods to optimize productivity in single dimensions are available. We found that to enhance heterologous production of a metabolite in a complex metabolic path requires the balancing of multiple levels of control (transcription, translation and activity), the design of metabolic modules and the use of data-driven models to guide further rounds of optimization. Hence, MHP is a workflow that has enabled the production of some terpenoids at yields previously not possible.

Zhao's COMPACTER is one useful method for optimizing metabolic pathway. Our MHP shares some similarity but yet distinct from the COMPACTER. Both approaches generate combinatorial libraries and the screening of desired phenotypes can be greatly assisted by high-throughput analytical methods. However, there is a clear difference between MHP and COMPACTER. COMPACTER approach has only one dimension of transcriptional control. In contrast, our approach involves three different dimensions - at transcriptional level controlled by promoters, at translational level controlled by 5'UTR or RBS and at enzyme level by enzyme screening or engineering. Controlling these biological events in a multidimensional manner is necessary to enable the high production of the desired metabolite. More significantly, with the COMPACTER approach, each gene is controlled by a few different promoter variants. This method is appropriate when optimizing a short biosynthetic pathway containing small number of genes, as is the case in their report where two genes (*cdt-1* and *gh1-1*) or three genes (*csXR*, *ctXDH* and *ppXKS*) were used. However, when used for studies requiring the control of longer biosynthetic pathways with more than 10 genes (such as in our study where we controlled a pathway of 14 genes), it will be rather challenging to screen a library with 4^{14} genotypes. To overcome this limitation, MHP introduced the modular concept of grouping multiple genes into single module, thus significantly reducing the total combinations and reducing library size from 4^{14} to 4^4 .

Salis's group pioneered the translational study by creating the useful webtool RBS Calculator. The group has published three reports regarding optimization of metabolic pathways including (1. *Molecular Systems Biology* (2014) 10: 731; 2. *Metabolic Engineering* (2015) 29:86–96); 3. *Microbial Cell Factories* (2016) 15:11). In the first study, they optimized a three-enzyme system (*crtE*, *crtB* and *crtI*) by using RBS Library Calculator. Similarly, in the second paper, they optimized a 5-enzyme pathway (synthetic Entner–Doudoroff pathway) by using only RBS library, without using a promoter library or enzyme variants. The third paper focused on how to obtain a predictive biophysical model from translation rates to protein expression

rather than optimizing a metabolic pathway. None of these studies used modular concepts as they merely optimized a relative short pathway (3-5 genes).

Voigt's group has published a few reports on how to design and optimize a metabolic pathway, and we have discussed their work and many other recent methods with ours in the Discussion section of our revised manuscript (line 324-336). One major difference is that MHP is not simply using the combination of promoters and RBSs but guided by a 'modular design' or standardization concept. Furthermore, MHP is a knowledge-based approach to enable the identification of optimal conditions rather than a purely screening approach that experimentally covers all the combinations.

A major premise of this work is the screening approach. Specifically, the underlying assumption is that extracellular concentration is proportional to intracellular production. However, this is not fully proven. In particular, while the authors show that there is different excretion for different carotenoids, it is not seen whether these values are constant and not a function of intracellular accumulation. This should be done at least in hindsight to prove the premise (based on the strains identified and observed). The issue of false negative and positive rates are not known and should be at least quantified to some extent.

We wish to thank the reviewer for this insightful suggestion. The use of extracellular astaxanthin as screening tool is one of novel aspect of this manuscript (its novelty is explained in Discussion section). To address the query on the underlying assumption, we have done an extensive study on the relationship between intracellular and extracellular carotenoids. First, we observed, for both positive and negative phenotypes, that the absorbance at 515 nm of culture media is highly correlated with the intracellular astaxanthin (but not other carotenoids) concentration (Fig. 6a). This observation is the basis for the use of medium absorbance as a screening indicator. The reason for choosing 515 nm is that zeaxanthin and β -carotene have lower absorption than astaxanthin, thus reducing their contributions to the absorbance. Furthermore, we extracted extracellular carotenoids by ethyl acetate and shown conclusively that the extracellular astaxanthin concentration is linearly correlated to the medium absorbance at 515 nm (Supplementary Fig 5). We also showed that the amounts of carotenoids accumulated extracellularly increases as their hydrophobicity decreases (Supplementary Fig. 10). As a result, astaxanthin has the highest transmembrane capability, and this might explain the linear correlation between intracellular and extracellular astaxanthin. We have now included the false positive results in Fig. 5b.

The meaning of error bars are not shown in the legends. Also, the number of replicates is not shown. This is not consistent with the quality control standards of Nature Communications.

The meaning of error bars have been incorporated into the figure legends.

In the abstract, the value of 320 mg/L is a titer, not a yield.

The error has been corrected.

The authors use the term "robust" to describe their MHP in most of the text, except for the title. The title is likely most correct. I am not seeing why / how this process is

robust. This is a random sampling approach with a built-in assumption of a hill climbing trajectory—which is not a robust algorithm for global optimization in many cases. This should be corrected.

As suggested by the reviewer, the term of 'robust' has been removed from the revised manuscript.

Reviewer #2 (Remarks to the Author):

The review is copied below. For ease of access to comments a Word document is also attached (containing identical information).

In the current manuscript, Zhang et al have developed a method they call 'Multidimensional Heuristic Process' for pathway optimisation. Before assessing this manuscript is necessary to provide background on multivariate approaches for combinatorial metabolic engineering. The first approach described in this family of approaches is multivariate modular metabolic engineering (MMME)¹. In this approach, enzymes are organized into operon modules comprising a set of pathway steps, and expression of each operon is varied simultaneously. Generally speaking, expression levels have been titrated in several ways, as noted in the current manuscript: gene copy number, promoter expression level, and ribosome binding site. Using this approach, an optimal flux balance between modules is identified. This optimality may arise from avoiding accumulation of intermediates which might be toxic and/or generally balancing flux through the pathway overall (it may be unclear why the particular expression balance is optimal). The multiplexed approach means that a very large sample space can be sampled as long as there is a facile and high-throughput screening technique. The EDASPO method developed out of the same lab² introduced a predictive element to the multivariate approach, allowing a much smaller solution space to be sampled to provide sufficient data resolution allowing prediction of module titration requirements that would allow further titre improvements.

In this work, the authors expand the approach to modulating both expression level (through screening libraries of regulatory elements, which includes promoters driving multigene operons as well as 5'UTRs to modulate expression of individual genes in each operon) combined with swapping specific enzyme variants from either natural sources or engineering products (so that specific activities/selectivities are also screened). These approaches (1) expanded the dimensions (and consequently the resolution and size of the solution space), improving the overall utility of the multivariate approach, and (2) allowed them to deal with a particular problem in the astaxanthin pathway, where two enzymes (crtZ and crtW) catalyse multiple different reactions each, and an appropriate balance of these reactions is required in order to optimise (maximise) flux through to the desired end product. These novel approaches and their impact are surprisingly not at all elucidated in the abstract, leaving the reader unsure of the significance of the study and indeed of the details of the method itself.

We wish to thank the reviewer for this excellent suggestion. The abstract has been re-written and the novelty of MHP is described in the revised manuscript.

The authors also developed a high-throughput assay allowing them to differentiate between the target compound, astaxanthin, and other red compounds (including lycopene, canthaxanthin). This was based on the observation that astaxanthin can traverse membranes far more readily than the carotenoid compounds, resulting in accumulation in the extracellular medium – providing a facile method to screen spectrophotometrically at the absorption maximum for astaxanthin. The best strain from the combinatorial engineering and screening strategy was then fermented in batch process bioreactor, with some process optimisation to improve growth rates, yielding impressive titres of 320 mg/L astaxanthin out of a total of 460 mg/L total carotenoids produced, with productivities exceeding the current industrial process. Finally, the broader utility of the MHP approach (using promoter and RBS combinations for three modules) was tested on two other isoprenoids, one sesquiterpene and one monoterpene.

Good results were obtained for the monoterpene product, but less impressive for the sesquiterpene (the lack of high throughput selection method limited execution of this component of the project).

Indeed, the sesquiterpene we obtained is not so impressive as compared to some reports in the literature. And this, to certain degree, reflects the limitation of MHP that its efficiency might be restricted when high-throughput analytical methods is not available for target products. However, this is not specific to our MHP, rather a general bottleneck of all the library screening based methods (such as mutagenesis, directed and in vivo evolution) in the field of metabolic engineering. To overcome this limitation, researchers have attempted to couple metabolite production with cell growth or fluorescence using transcription-based biosensors (Biotechnol Adv. 2017 Dec;35(8):950-970). However, there is only limited biosensors for a small portion of growth-related metabolites, for most of metabolites, there are no suitable sensors and it is challenging to develop sensors with high sensitivity. In addition, better than evolution/mutagenesis methods, MHP has the flexibility to choose the regulatory elements (promoters or 5'UTRs). For those universal molecules, we can reduce the total combinations by using a small library (e.g. with two instead of four promoters). With these data, we can build a data-driven model to guide the second round optimization. This concept has been included in Fig 4b. In fact, this is one of the directions we are currently working on. The discussion has now been added to the Discussion section (Line 400-406).

The manuscript suffers from numerous communication problems, in particular around style, use of language and grammar, and structure. This makes it a bit impenetrable in places, and the overall significance of the story might easily be lost as the context and background is not clearly explained in the introduction. Notwithstanding this, the manuscript is an original contribution and demonstrates a remarkably effective approach at increasing production levels for the target compound, astaxanthin. This is likely to be generalizable, in particular for target compounds where a facile high-throughput assay is available: less compelling results are presented for the other isoprenoid based target compounds, but the method is far less robustly applied in those cases and no HTP screening method exists, significantly curtailing the ability to effectively identify high producers. The study is ambitious in theoretical scale (although less so in execution). Collection of a bit more data (as outlined below) would make the paper more robust and would likely increase the scope and impact of the findings – specifically, the authors are using

titrations of transcription, translation, and enzyme activity – but they are measuring none of these directly – the only measure the metabolites. The only expression level data presented is reporter gene activity, which doesn't necessarily scale directly, and they use theoretical expression strengths also in places. Unsurprisingly, correlations are not identified between these data and the metabolite accumulation data. For completion, it would be appropriate to pay more attention to quantitative measurement at all stages of expression to allow meaningful correlative analysis with metabolite data.

The quantitative data for both mRNA and enzymes would be very useful to obtain a predictive model or framework for the metabolic pathways. This knowledge-building process is one aspect of MHP as well (Fig. 4b). We have now included the mRNA measurement of *crtY1* gene, and its mRNA level correlated well with the fluorescence of the reporter protein eGFP (Supplementary Fig. 1). In addition, we also compared the mRNA levels for the twelve strains in Fig. 3c. As expected, all the strains have similar mRNA copies of *crtZ* or *crtW* (Supplementary Fig. 10) as they were controlled by the same vectors and promoters. However, due to technical limitations, we could not obtain quantitative data for protein levels of these enzymes. This is simply due to the low expression of the proteins as we are not expressing all the fourteen enzymes to sufficiently high levels to be quantified. It is well known that high-level expression of enzymes negatively affects production due to metabolic burden. Hence, we used low copy plasmids and tune down the promoter strength for many modules. Furthermore, as these enzymes are native forms without any tags, we were unable to detect enzymes by western bolt.

It is likely that this multi-dimensional approach of titrating at various levels of expression and enzyme activity will find broader utility in metabolic engineering for different products, especially where high throughput analysis for the target product is available.

We wish to thank the reviewer for this encouraging comment.

General

1. English use and grammar needs to be improved throughout the text.

We have engaged a native English speaking scientist/administrator to assist with the editorial.

2. Communication could be significantly improved by working on the text in the introduction. Such improvement would place the results section in a much better context. A proper description of the original multivariate approach as well as all of the subsequent improvements and modifications to the overall theme should be done in the introduction as part of the literature review.

As suggested, to make it clearer and more fluid, we have re-written the abstract and re-organized the introduction. The background regarding multivariate modular approach is included in the introduction section (Line 62-68). The background regarding the experimental design-aided systematic pathway optimization (EDASPO) method is also included in the introduction section (Line 69-75). The improvement of MHP over existing multivariate methods is highlighted as well (Line 77-96).

3. A number of breaches of scientific writing protocol are made (e.g., presenting data from the results as part of the introduction)

All the mentioned issues have now been corrected. The data have been moved from Introduction to Results.

4. It's a pity that strains from the library were discarded in favour of focussing strongly on astaxanthin producing strains (page 7). A far more detailed picture of the expression balance requirements for different intermediates and end-products might have been obtained by characterising and examining a larger range of strains.

In order to address this interesting suggestion, we have re-picked a few more red colonies and analyzed the false negative data (Line 188-195). We then supplemented the false negative data into Fig. 5 b, c and d. As all of the false negative phenotypes had a relatively low production of astaxanthin, the inclusion of false negative data did not improve the predictability between astaxanthin titers and the translational rates. The correlation is still weak between astaxanthin titers and the translational rates, presumably due to yet to be characterized post-translational or regulatory effects.

5. A lot of assumptions are made about relationships between translation rates and enzyme activities (e.g. Fig 4c and other statements in the text); however, the actual transcription/translation is never measured – assumptions are made based on theoretical transcription levels and reporter gene data. This is problematic for the current study, since it is describing and attempting to parameterise the approach. The manuscript would be improved by measuring transcription and ideally also translation quantitatively. This should be feasible in a subset of useful strains, as the refined sets are relatively small. Such analyses might reveal substantially more about the driving factors behind the observed phenomena.

This is the same suggestion raised above. The quantitative data for both mRNA and enzymes would be very useful to obtain a predictive model or framework for the metabolic pathways. This knowledge-building process is actually one aspect of MHP (Fig. 4b). We have now included mRNA measurement of *crtY1* gene, and its mRNA level which correlated well with the calculated level based on fluorescence of the reporter protein eGFP (Supplementary Fig. 1) and also with its product β -carotene (Fig. 2g). In addition, we also compared the mRNA levels for the twelve strains in Fig. 3c. As expected, all the strains have similar mRNA copies of *crtZ* or *crtW* (Supplementary Fig. 10) as the same promoters controlled them. As discussed above, protein levels of the corresponding enzymes were not measured as the expression levels were below the detection limit.

6. No growth data are presented in the results. This is a significant oversight, since pathway imbalances commonly cause growth defects. A difference in growth rate might significantly affect the observed carotenoid accumulation/titre as a given time point. On the other hand, this can be standardised for by presenting ppm as is done frequently. Notwithstanding this, growth profiles can be quickly assessed by microtitre growth plate readers. This should be done as the growth profile is a standard part of any metabolic engineering strategy and may shed more light on the physiology and phenotypes. Indeed, the authors only briefly mention growth defects as a result of induction of the pathway genes in the bioprocess section. This confirms the importance of gathering more growth data throughout this work.

All the growth data have now been included in the figures, including Fig 2, 3, 5, 6 and 8. In addition, for better comparison of intracellular and extracellular concentration and for the sake of clarity, we have replaced all the intracellular contents (ppm) with titers (mg/L) in all the figures.

7. In the results I find it a bit painful to move between ppm and titre. In general, titre is the more commonly-reported result.

As suggested, for better comparison of intracellular and extracellular concentration, we have replaced all the intracellular contents (ppm) with titers (mg/L) in all the figures.

8. Statistical support for some of the comments made is not provided.

The meaning of error bars and statistical significance have been included in the corresponding figure legends.

9. A limitation in the execution of this protocol is the need for a high throughput screen to effectively exploit the approach. This is evident in the relative success for the different products targeted. This limitation should be properly explained and explored in the discussion, and for completeness, it should be referred to in the abstract.

This is the same suggestion raised above. Due to the time consuming and high cost of operating mass spectrometry, it is only possible to analyze a limited number of combinations. In the case of nerolidol, a colorless metabolite, the production was lower than the highest report and this was likely to be due to the less comprehensive survey of the number of combinations. We agree with the reviewer that high-throughput readouts lead to better performance of MHP. The discussion has now been incorporated to the Discussion section (Line 401-406).

Abstract

10. From the abstract, I have absolutely no idea what the 'multidimensional heuristic process' is – despite it being the title of the paper. A description of what the process is should be embedded in the abstract since it's the central focus of the paper. The product itself is almost incidental. One should be able to determine from the abstract what the novelty of the story is, and it's impossible to do that from the current abstract.

We wish to thank the reviewer for this suggestion. The abstract has now been modified accordingly.

11. It's always a bit suspicious when titres are reported for one compound but only fold improvement for others (astaxanthin vs. nerolidol and linalool)

This has now been rectified to report all production in terms of titres.

Introduction

12. From a biotechnology perspective, it should probably be noted that astaxanthin is approved as an oral supplement (despite having very low oral absorption) and as a food colourant and feed additive.

We have incorporated this piece of information into the Introduction section (Line 37).

13. Lines 53-56: the logical gap here is that the previously engineered strains presumably accumulate a large amount of beta-carotene as a result of upstream pathway engineering, but that this is not converted efficiently into astaxanthin (despite trying several different ketolases and hydroxylases). There is context missing in this section, as the authors focus in this paragraph on (a) sufficient production of lycopene, or (b) problems with crtY, crtW, and crtZ. There is plenty of information available in the literature on accumulation of products that are intermediates in this pathway (primarily lycopene and beta-carotene, for which reasonably high titres can be achieved).

This part has been revised for clarity (line 52-60).

14. Line 64: when multivariate modular metabolic engineering is first mentioned, it would be useful to explain what it is and what its utility is before pointing out the restriction of the method and then introducing the modified EDASPO method

The background about multivariate modular metabolic engineering has now been described in Line 64-68.

15. The EDASPO method, and in particular the differences of this method compared to the MMME method, should be described (not just the advantages, but the methodological improvement); and the acronym should be defined at first use

The background about experimental design-aided systematic pathway optimization (EDASPO) and its difference with MMME has now been described in Line 69-75.

16. It is unclear from the context of the text whether the data from the EDASPO method for lycopene and beta-carotene presented on line 68 was published previously (in which case it should be cited both here in the text and in the figure caption, and I do not see a need to re-publish the data) or whether it was generated for this work (in which case it should be in the Results).

The data were not published previously. We have now incorporated this set of data into the section on Results (Fig. 2).

17. Similarly the data in the results should not be referred to in the Introduction.

We have moved all the data from Introduction to the Results section.

18. Lines 74-75: 'In dealing with complex pathways, previous studies have focused on using either one or two of the controlling dimensions (gene copy 24, 26, promoter 24, 25, 27, 28 or ribosome binding site (RBS) 29, 30), which leaves either local or global pathway not sufficiently optimized and hence limiting the final yields'. It is

unclear what the authors mean by this and why sufficient optimisation is not obtained using these approaches. Thus, the overall significance of the work is unclear.

We have re-written the Introduction section. Now the significance, definition and differences from other methods have been underlined in Line 76-96.

Results

19. Lycopene was used as a screen in several places. It is worth keeping consideration of the fact that lycopene is not always a direct readout of pathway flux, since lycopene is highly sensitive to oxidation and cellular stress results in production of reactive oxygen species. This can affect lycopene titres as the oxidised products are not red³

Indeed, the report (Microb Cell Fact 2015, 14 (1), 1-16) systematically studied the mechanism of lycopene oxidation and concluded that it is not an ideal metabolite indicator. In line with this argument, we are aware of this sensitivity issue when using lycopene as an indicator to optimize the isoprenoid pathway. However, as we only selected the colonies with the most intense color, the relative amounts of lycopene accumulated will predictably be higher than those which are pale in color. This is consistent with the findings of our observation (Fig. 2c). In addition, unlike the previous studies (Proc. Natl. Acad. Sci. U. S. A. 107, 13654-13659 ; Biotechnol. Bioeng. 113, 2661-2669) where lycopene was used only as a surrogate reporter to optimize the production of other **non-carotenoid** terpenoids, we selected high lycopene producers to directly produce carotenoids β -carotene and astaxanthin (lycopene is their common precursor).

20. It is unclear what the arrows on Figure 2b mean from the caption. You have to guess from the text on line 104

The arrows indicated the comparison of the results from four different combination of enzymes, and a description has now been incorporated into the legend of Figure 3 (previously as Figure 2).

21. Line 97: it's unclear why Fig 3a is referred to here

The reference to Figure 3a has now been removed.

22. Line 105: the authors state that strain 22 had the highest astaxanthin, but this is incorrect based on figure 2b. Strain 2'2 clearly has much higher levels of astaxanthin (though decreased overall flux to the carotenoid family, and restricted product profile – it has 93 % astaxanthin and only a small amount of lycopene. This strain therefore is very interesting since it has the potential to produce a high level of specific product, and I am confused as to why this strain was not further investigated.

Indeed, strain 2'2 was the best strain shown in Fig. 3c (previously shown as Figure 2b). But in line 105 (previous submission), the context is that we were comparing 4 different combination of enzymes, 11, 12, 21 and 22 (which are pointed out by arrows in Fig. 3), and among the 4 combinations, 22 is the best combination. Following that, we further compared RBS effect on different astaxanthin production, and as mentioned in line 168-172, strain 2'2 was a better strain than 22, with 93% astaxanthin purity.

Strain 22 and strain 2'2 have the same set of enzymes, crtW2 and crtZ2. The only difference is that they have two different RBS controlling crtZ2 expression. From this result, we knew that RBSs are critical to improve the yield and purity of astaxanthin. Hence, in the following section, we focused on the same set of enzymes but did a more thorough fine-tuning of RBS controlling crtZ2 and crtW2 expressions (their RBS sequences were in Supplementary Table 2, 3 and 4). Because of the RBS of crtZ2 was re-adjusted with a synthetic RBS library (Supplementary Table 3 and 4 and Supplementary Fig. 3 in the revised file), it was different from both strain 22 and strain 2'2. Strain 2'2 has a good purity but its yield is relatively lower. And strain E (Fig. 6b), identified with medium screening, has both higher yield and purity, hence we used it in the following fermentation processes.

23. Line 108-110: 'Consistent with the hypothesis, the relative expressions of ketolase and hydroxylase have significant effects on the production of astaxanthin and total carotenoids (Fig. 2b).' The statement does not align with the cross-reference to the figure, which only shows the carotenoid accumulation data and not the relative expression levels (which I think are deduced from the GFP expression data shown in suppl. figure 1a, though that is not clear from the text, and this is a poor way to do the assessment – expression levels can be easily measured using RT-PCR and compared with the product levels in the relevant strains to determine if a correlation exists. And in this case, a proper statistical analysis should be done. Like explained previously, we have measured the transcription levels of crtZ and crtW of the twelve strains in Fig. 3c (previously as Fig. 2b). The difference of mRNA copies of these were not significantly different (Supplementary Fig. 10). However, we could not accurately quantify enzyme amounts, due to technical limitation where the protein levels were below the detection limit. In order to clarify this issue in the text, we have now changed the sentence to “the choices of RBSs of ketolase and hydroxylase have significant effects on the production of astaxanthin and total carotenoids”.

24. Please convert ppm to titre for key strains (both can be included).
All the 'ppm' data have been replaced with 'titre' data.

25. The sample size for the sesquiterpene and monoterpene study was small compared to the theoretical sample size, but this is understandable considering that there is no high throughput screen for these products
The discussion about this limitation with MHP has been incorporate in line 401-406.

Discussion

26. The fold increases for the nerolidol/linalool work are not bad, and the linalool titres are very good. But the final titres of nerolidol are not impressive compared to the best strains currently published (and in general for a sesquiterpene). It would be polite to acknowledge that, and note the studies which have produced high titres of these compounds to date and why they have been more successful (this might suggest a next step engineering approach that the authors could include in future studies applying the MHP for this target compound. Similarly, an analysis of the

engineering steps required to improve monoterpene titer might shed insights on future steps for that target)

The discussion on the titers and potential future work on nerolidol and linalool has been now incorporated in line 391-406.

27. Lines 284-288: carotenoids have been produced at least an order of magnitude higher than the levels reported here for astaxanthin. It will probably be a while before astaxanthin cell titers reach levels that require engineering of export components. Furthermore, as noted, this is one carotenoid which is reasonably effectively exported from the cell. So I cannot see that engineering transporters are the next step. Instead, identification of other bottlenecks for conversion through the pathway would be required.

We agree that further optimization of the pathway could lead to an even higher astaxanthin titer. However, engineering an efflux transporter may yet be another strategy to increase the titer. The rationale is as follows. With the best strain E, there was less accumulation of other carotenoids, as astaxanthin content is ~83%. Therefore, it is likely that other factors may be limiting the astaxanthin production, such as cellular storage capability and toxicity of astaxanthin. In these situations, the use of efflux transporters might mitigate these challenges and potentially improve the production rate and final titers. This is similar to the ideas in the reports cited by us in the previous version of the manuscript and mentioned by reviewer 3 (Proc. Natl. Acad. Sci. U. S. A. 110, 7642-7647 and Appl. Microbiol. Biotechnol. 100, 869-877). In addition, if there are transporter/s preferentially exporting astaxanthin, then the purity of the extracellular astaxanthin will be further improved.

28. It is appropriate to highlight that better in the discussion that the method is constrained somewhat where there is a lack of high throughput selection method, as this is a limitation of the approach.

A discussion on the limitation of MHP due to the lack of a reliable selection method has now been incorporated into the Discussion section (Line 401-406).

Materials and methods

29. The cultivation method for terpenoid production is missing. Noteworthy is the fact that these C10 and C15 terpenoids are highly volatile and cultures are typically performed in sealed flasks using an organic phase to sequester the terpenoids to minimize loss. Since an organic phase sample was used for the analytics, I presume that the two-phase cultivation was performed, but the details for this component must be described.

The missing cultivation method has now been described in line 493-497.

References cited

1. Ajikumar, P. K.; Xiao, W.-H.; Tyo, K. E. J.; Wang, Y.; Simeon, F.; Leonard, E.; Mucha, O.; Phon, T. H.; Pfeifer, B.; Stephanopoulos, G., Isoprenoid pathway optimization for taxol precursor overproduction in *Escherichia coli*. *Science* 2010, 330 (6000), 70-74.
2. Zhang, C.; Zou, R.; Chen, X.; Stephanopoulos, G.; Too, H. P., Experimental design-aided systematic pathway optimization of glucose uptake and deoxyxylulose

phosphate pathway for improved amorphaadiene production. Appl Microbiol Biotechnol 2015, 99 (9), 3825-37.

3. Bongers, M.; Chrysanthopoulos, P. K.; Behrendorff, J. B. Y. H.; Hodson, M. P.; Vickers, C. E.; Nielsen, L. K., Systems analysis of methylerythritol-phosphate pathway flux in E. coli: insights into the role of oxidative stress and the validity of lycopene as an isoprenoid reporter metabolite. Microb Cell Fact 2015, 14 (1), 1-16.

Reviewer #3 (Remarks to the Author):

This manuscript describes the improvement of natural products biosynthesis in E. coli through a so-called multidimensional heuristic process (MHP), which combines optimization of promoters, RBS and enzyme source for pathway enzymes. Although high productivity and yield of natural products such as astaxanthin was achieved using this process, the novelty and technical advance of the work seem to be insufficient to meet the publication criteria of Nature Communications. The difference between the reported method and the previously reported modular pathway optimization approaches is not big enough to distinguish it as an inspiringly new method.

This is the similar comment by the first reviewer. We have addressed this above. We have now revised the Introduction (Line 64-96) and Discussion (Line 322-352) sections to explain the novelty and differences between MHP and other existing pathway engineering approaches. Briefly, there are two major features. Firstly, MHP is a modular engineering approach by grouping multiple genes into single module thus significantly reducing the total possible combinations. Taking astaxanthin as an example, we were simultaneously controlling 14 enzymes with four different promoters. Modularization significantly reduced the total combination from 4^{14} to 4^4 . Secondly, we explored multiple dimensions (transcription, translation and enzymes) to control the metabolic pathways. The first dimension involved promoters that can globally control the transcription of the whole pathway. The second dimension was the use of variable 5' untranslated regions (5'UTR) or RBS to generate variability in the expression of the enzymes in specific module. The third dimension was the use of enzymes with better activities or selectivities. This local dimension is to overcome the imbalance of enzymes in specific module, thus is an important supplementation to the transcription-based global pathway optimization.

Some more issues are listed below:

1. The manuscript is not well structured and the findings are insufficiently discussed. In particular, the technical details are not well presented.

The manuscript has been re-structured following the suggestions of the second reviewer. Technical details have now been incorporated as well.

2. Although the authors emphasize astaxanthin synthetic pathway as challenging in pathway optimization since it is a long pathway involving 15 reactions, the experiments described in this paper are merely combination of RBS and enzyme sources (two for each enzyme) for CrtW and CrtZ, whereas the more labor-intensive promoter optimization for multiple pathway enzymes carried out using the previously reported EDASPO method was not described in details. Especially, for the abbreviation EDASPO, even the full name was not mentioned. In addition, the failure of EDASPO method in production of high astaxanthin was attributed to its inability to

deal with the complex biosynthetic pathway, without further explanation and discussion on the possible synergy among different dimensions of regulatory factors. We have re-written the Introduction. The background on experimental design-aided systematic pathway optimization (EDASPO) and its difference with MMME has now been discussed in Line 67-78. The limitation of EDASPO has been described in Line 75-81. In addition, the results with EDASPO method has now been incorporated into the Result section.

3. The two-phase fermentation process developed for improved astaxanthin production is inspiring for carotenoids biosynthesis, but the optimization of the organic solvent layer may be introduced in more details, and more discussion may be included on extracellular secretion of carotenoids and two-phase fermentation of lipophilic products. Literatures such as “Appl Microbiol Biotechnol (2016) 100:869–877” and “Yeast 2010; 27: 983–998” may be referred to.

Thank you for pointing this out. We have now provided a discussion about the two-phase fermentation (Line 382-388). The concept of two-phase fermentation has been previously demonstrated for the production of extracellular carotenoids with expression of efflux transporters (Appl Microbiol Biotechnol (2016) 100:869–877). However, their overall titers were relatively low (<10 mg/L) and they tested only in small tubes but not in bioreactors, thus may not be industrially relevant. Nevertheless, these studies (*Proc. Natl. Acad. Sci. U. S. A.* **110**, 7642-7647 (2013); Appl Microbiol Biotechnol (2016) 100:869–877; Yeast 2010; 27: 983–998;) reinforced the idea that engineering efflux transporters could be a future direction to further improve the production of carotenoids.

4. For astaxanthin biosynthesis in engineered microorganisms, a recent publication on *Kluyveromyces marxianus* might be included (Lin Y J, Chang J J, Lin H Y, et al. Metabolic Engineering a Yeast to Produce Astaxanthin. Bioresource Technology, 2017. doi.org/10.1016/j.biortech.2017.07.116)

We wish to thank the reviewer for highlighting these publications. In fact, the paper was originally cited in the Supplementary Table 1 (now as Supplementary Table 5 in revised file) as reference 14.

Reviewers' Comments:

Reviewer #1 (Remarks to the Author):

The authors have made improvements in their manuscript, but still do not fully address their work and method in light of others in the field. The authors have chosen a rather narrow view of the field as a whole. I had mentioned that similar factorial studies have been conducted. Take for example DOI: doi: 10.1093/nar/gkv1071 In this paper, a multi-variate optimization and factorial design was conducted. Papers like these need to be addressed to show the differentiating factors of this work. There are several others in the field as well and the authors should likewise address those.

It appears a rather high false positive rate was seen in this work (17 / 36 colonies were negative). To what is this attributed? More details about robustness are needed here.

Reviewer #2 (Remarks to the Author):

I find the revised manuscript very much better than the original version. The Abstract still needs some work (outlined below), but the Introduction and Results are vastly improved. Some grammar editing is still required, in particular, standardisation of tense (abstract is written in present tense – very unusual in scientific writing; it should be past tense)[ED: the reviewer is incorrect in this instance; abstract should be written in present tense.] ; but this can probably be performed at copyediting phase. A few other minor issues were identified, but overall the manuscript is reasonable.

1

The Abstract still fails to properly outline the novelty of the work relative to previous work. Specifically, as noted previously, the authors' approaches (1) expanded the dimensions (and consequently the resolution and size of the solution space), improving the overall utility of the multivariate approach, and (2) allowed them to deal with a particular problem in the astaxanthin pathway, where two enzymes (crtZ and crtW) catalyse multiple different reactions each, and an appropriate balance of these reactions is required in order to optimise (maximise) flux through to the desired end product. Specific reference to the novelty with respect to previous multivariate approaches is necessary to inform the reader of the significance of the work. The authors themselves have done a very nice summary of the novelty in the Discussion section:

323: '...Modular pathway optimization represents a
324 powerful strategy to tackle this issue with distinctive advantages (e.g., isolating the
325 genes from the regulatory network by modularization, facilitating more efficient
326 construction and screening of combinatorial libraries by reducing total combinations)

327 than semirational approaches 36, 37. As genes in the same module are controlled
328 simultaneously, it is best to categorize enzymes of similar turnovers into the same
329 module. However, in vivo enzymatic kinetics are difficult to determine and existing
330 modular approaches do not take this issue into consideration 23, 24. MHP addresses this
331 issue by introducing global and local optimization with three-dimensional elements
332 (transcription/ translation/ enzyme). Global optimization is the same as existing modular
333 approaches by using promoters to balance all the modules along the pathway. The
334 imbalance within a module is then overcome by local optimization (translational control
335 by 5'UTRs or RBSs, and enzyme variants). The feature also makes MHP different from
336 existing fractional design-based pathway optimization methods 24, 38, 39.

I recommend summarising this text as part of the Abstract to properly highlight the significance of their work.

2

There is an error in the Discussion in the following section:

“393 The overall lower yields of monoterpenoids than
394 sesquiterpenoids was likely to be due to the lower expression/activity of the plant GPP
395 synthase in *E. coli* than the native FPP synthase.”

The lower monoterpenoid production can be explained by the poor kinetic properties of the GPP synthase, but not the lower sesquiterpenoid production.

3

There is an error in the Results section, lines 495-497:

“Specifically, fresh cells (20 μ l) were inoculated into 1 ml fresh media in 14 ml BD Falcon™ tube and covered with 200 ml dodecane (with 100 mg 1-1 β -caryophyllene as internal control).”

Presumably, the authors meant ‘200 μ l’ not ‘200 ml’. It's also unwise to use a tube with such a large headspace relative to culture volume when working with a volatile product - this will unnecessarily increase the impact of volatilisation, since not all volatiles are captured by the dodecane layer. Also, please add the time period used for incubation of the cultures in this tube. For future reference, glass fermentation vessels sealed with an inert lid (e.g. silicon or teflon lined) should be used for volatiles analysis. Over time, the volatiles will dissolve into plastic.

Reviewer #3 (Remarks to the Author):

The manuscript has been significantly improved after revision. All my concerns have been well addressed.

Responses to Reviewers:

We wish to thank the reviewers again for their insightful comments and invaluable suggestions. We have revised the manuscript accordingly especially the abstract and the background of existing multivariate metabolic engineering methods. The point-by-point response to the reviewers' comments are included below.

REVIEWERS' COMMENTS:

Reviewer #1 (Remarks to the Author):

The authors have made improvements in their manuscript, but still do not fully address their work and method in light of others in the field. The authors have chosen a rather narrow view of the field as a whole. I had mentioned that similar factorial studies have been conducted. Take for example DOI: doi: 10.1093/nar/gkv1071 In this paper, a multi-variate optimization and factorial design was conducted. Papers like these need to be addressed to show the differentiating factors of this work. There are several others in the field as well and the authors should likewise address those.

As suggested by the reviewer suggested, we have now incorporated these multi-variate methods in the Introduction section (Line 79-81, the revised manuscript). We have attempted to include as broad a coverage as possible. However, due to the restriction of manuscript length and total number of references, we could only choose one or two reprehensive examples for similar methods (Line 63-81, the revised manuscript). In essence we have cited references discussing related methods such as that suggested by the reviewer.

It appears a rather high false positive rate was seen in this work (17 / 36 colonies were negative). To what is this attributed? More details about robustness are needed here.

As we have mentioned in the Results section 3.3 and Discussion section, two likely factors contributed to the high false positive rate of colony-colour based screening. Firstly, the confounding effect of accumulation of other red carotenoids such as lycopene (ZW2, W4 and ZW15) or canthaxanthin (ZW9 and Z6, Fig. 5b). The other factor was the subjective judgement of operators based on color. Hence, the high false positive rates. In order to mitigate this issue, an objective and quantitative method was developed by measuring the absorbance in the cell media (see Result section 3.3).

Reviewer #2 (Remarks to the Author):

I find the revised manuscript very much better than the original version. The Abstract still needs some work (outlined below), but the Introduction and Results are vastly improved. Some grammar editing is still required, in particular, standardisation of tense (abstract is written in present tense – very unusual in scientific writing; is should be past tense)[ED: the reviewer is incorrect in this instance; abstract should be written in present tense.] ; but this can probably be performed at copyediting phase. A few other minor issues were identified, but overall the manuscript is reasonable.

In accordance to the requirement of “Nature communications”, the abstract is written in present tense.

1

The Abstract still fails to properly outline the novelty of the work relative to previous work. Specifically, as noted previously, the authors’ approaches (1) expanded the dimensions (and consequently the resolution and size of the solution space), improving the overall utility of the multivariate approach, and (2) allowed them to deal with a particular problem in the astaxanthin pathway, where two enzymes (crtZ and crtW) catalyse multiple different reactions each, and an appropriate balance of these reactions is required in order to optimise (maximise) flux through to the desired end product. Specific reference to the novelty with respect to previous multivariate approaches is necessary to inform the reader of the significance of the work. The authors themselves have done a very nice summary of the novelty in the Discussion section:

323: ‘...Modular pathway optimization represents a
324 powerful strategy to tackle this issue with distinctive advantages (e.g., isolating the
325 genes from the regulatory network by modularization, facilitating more efficient
326 construction and screening of combinatorial libraries by reducing total
combinations)
327 than semirational approaches 36, 37. As genes in the same module are
controlled
328 simultaneously, it is best to categorize enzymes of similar turnovers into the
same
329 module. However, in vivo enzymatic kinetics are difficult to determine and
existing
330 modular approaches do not take this issue into consideration 23, 24. MHP
addresses this
331 issue by introducing global and local optimization with three-dimensional
elements
332 (transcription/ translation/ enzyme). Global optimization is the same as existing
modular
333 approaches by using promoters to balance all the modules along the pathway.
The
334 imbalance within a module is then overcome by local optimization (translational
control
335 by 5’UTRs or RBSs, and enzyme variants). The feature also makes MHP
different from
336 existing fractional design-based pathway optimization methods 24, 38, 39.

I recommend summarising this text as part of the Abstract to properly highlight the significance of their work.

Based on the reviewer’s comments, we have rewritten the abstract to highlight the novelty of and utility of this approach with respect to previous multivariate methods and presented examples of producing astaxanthin and other molecules.

2

There is an error in the Discussion in the following section:

“393 The overall lower yields of monoterpenoids than
394 sesquiterpenoids was likely to be due to the lower expression/activity of the
plant GPP
395 synthase in *E. coli* than the native FPP synthase.”

The lower monoterpenoid production can be explained by the poor kinetic properties of the GPP synthase, but not the lower sesquiterpenoid production.

We wish to thank the reviewer for pointing these errors and we have rectified accordingly.

3

There is an error in the Results section, lines 495-497:

“Specifically, fresh cells (20 μ l) were inoculated into 1 ml fresh media in 14 ml BD Falcon™ tube and covered with 200 ml dodecane (with 100 mg l⁻¹ β -caryophyllene as internal control).”

Presumably, the authors meant ‘200 μ l’ not ‘200 ml’. It’s also unwise to use a tube with such a large headspace relative to culture volume when working with a volatile product - this will unnecessarily increase the impact of volatilisation, since not all volatiles are captured by the dodecane layer. Also, please add the time period used for incubation of the cultures in this tube. For future reference, glass fermentation vessels sealed with an inert lid (e.g. silicon or teflon lined) should be used for volatiles analysis. Over time, the volatiles will dissolve into plastic.

We wish to thank the reviewer for suggesting the use of ‘closed systems’ to reduce loss of volatile products. This will be the subject of our next study and should improve the yield even more than reported herein.

The typo error of ‘200ml’ has been rectified by replacing with ‘200 \$\mu\$ l’.

Reviewer #3 (Remarks to the Author):

The manuscript has been significantly improved after revision. All my concerns have been well addressed.